EMBO
Molecular Medicine

# In vitro and in vivo inhibition of the host TRPC4 channel attenuates Zika virus infection

Xingjuan Chen [1,2,8], Yunzheng Yan [2,8], Zhiqiang Liu [3,8], Shaokang Yang [2,8], Wei Li [2], Zhuang Wang [1], Mengyuan Wang [1], Juan Guo [1], Zhenyang Li [2], Weiyan Zhu [2], Jingjing Yang [2,4], Jiye Yin [2], Qingsong Dai [2], Yuexiang Li [2], Cui Wang [3], Lei Zhao [2], Xiaotong Yang [2], Xiaojia Guo [2], Ling Leng [5], Jiaxi Xu [6], Alexander G Obukhov [7✉], Ruiyuan Cao [2✉] & Wu Zhong [2✉]

## Abstract

Zika virus (ZIKV) infection may lead to severe neurological consequences, including seizures, and early infancy death. However, the involved mechanisms are still largely unknown. TRPC channels play an important role in regulating nervous system excitability and are implicated in seizure development. We investigated whether TRPCs might be involved in the pathogenesis of ZIKV infection. We found that ZIKV infection increases TRPC4 expression in host cells via the interaction between the ZIKV-NS3 protein and CaMKII, enhancing TRPC4-mediated calcium influx. Pharmacological inhibition of CaMKII decreased both pCREB and TRPC4 protein levels, whereas the suppression of either TRPC4 or CaMKII improved the survival rate of ZIKV-infected cells and reduced viral protein production, likely by impeding the replication phase of the viral life cycle. TRPC4 or CaMKII inhibitors also reduced seizures and increased the survival of ZIKV-infected neonatal mice and blocked the spread of ZIKV in brain organoids derived from human-induced pluripotent stem cells. These findings suggest that targeting CaMKII or TRPC4 may offer a promising approach for developing novel anti-ZIKV therapies, capable of preventing ZIKV-associated seizures and death.

**Keywords** Zika Virus; TRPC4 Channel; Calcium; Epilepsy; Antiviral Target
**Subject Category** Microbiology, Virology & Host Pathogen Interaction

## Introduction

Zika virus (ZIKV) is a 10.7 kb positive-stranded RNA virus, which belongs to the *Flavivirus* genus of the *Flaviviridae* family and is predominantly transmitted to humans via the bites of infected *Aedes aegypti* and *Aedes albopictus* mosquitos (Bueno et al, 2016). Sexual contact, mother-to-fetus, and blood transfusion-related transmissions are also documented as possible routes for ZIKV spread in the human population (Barjas-Castro et al, 2016; Mansuy et al, 2016). The recent ZIKV outbreaks have revealed new ZIKV variants exhibiting an increased infectivity, calling attention to the danger of future epidemics (Aubry et al, 2021; Fajardo et al, 2016; Gurung et al, 2020).

ZIKV is highly neurotropic and is especially harmful for the immature nervous system. ZIKV infection has been linked to Guillain–Barré syndrome (GBS) in adults and is strongly associated with the development of severe fetal brain abnormalities, such as hydranencephaly and microcephaly (Barjas-Castro et al, 2016; Moura da Silva et al, 2016; Sarno et al, 2016; Ventura et al, 2016). Remarkably, the infected infants with microcephaly are reported to present with recurrent seizures (Asadi-Pooya, 2016; Carvalho et al, 2020), and 60% of normocephalic babies exposed to ZIKV in utero were also documented to suffer from seizures (Asadi-Pooya, 2016; Carvalho et al, 2017). Moreover, there are recent alarming reports indicating that Congenital Zika Syndrome infants, born to mothers infected with ZIKV during the first trimester of pregnancy, present with a greater risk of epilepsy and death in early infancy compared to the non-infected infants (Oliveira-Filho et al, 2018; Souza et al, 2021). Thus, the consequences of ZIKV infection may be severe, especially in infants. Therefore, it is important to fill the gaps in our knowledge on the mechanisms underlying ZIKV infection and ZIKV-related neurological complications.

$Ca^{2+}$ is an important second messenger in mammalian cells, which modulates various cellular functions ranging from stress response, synaptic plasticity, to endosome formation. Host cell dysfunction following infection with a virus, including ZIKV, is accompanied by an increased intracellular $Ca^{2+}$ concentration (Chen et al, 2019). ZIKV can hijack the host's intracellular $Ca^{2+}$ machinery to favor its replication, and this may lead to an array of host disease (Chen et al, 2019). The major pathway which ZIKV employs to enter the host cells involves clathrin-mediated

[1]Institute of Medical Research, Northwestern Polytechnical University, 710072 Xi'an, Shanxi, China. [2]National Engineering Research Center for the Emergency Drug, Beijing Institute of Pharmacology and Toxicology, Beijing, China. [3]Beijing Institute of Basic Medical Sciences, Beijing, China. [4]School of Pharmaceutical Sciences, Hainan University, Haikou, China. [5]State Key Laboratory of Complex Severe and Rare Diseases, Peking Union Medical College Hospital, Chinese Academy of Medical Sciences and Peking Union Medical College, Beijing, China. [6]Department of Physiology and Pathophysiology, Xi'an Jiaotong University Health Science Center, 710061 Xi'an, Shanxi, China. [7]The Department of Anatomy, Cell Biology & Physiology, Indiana University School of Medicine, Indianapolis, IN 46202, USA. [8]These authors contributed equally: Xingjuan Chen, Yunzheng Yan, Zhiqiang Liu, Shaokang Yang. ✉E-mail: aobukhov@iu.edu; caoruiyuan@bmi.ac.cn; zhongwu@bmi.ac.cn

endocytosis (Agrelli et al, 2019), which is a $Ca^{2+}$-dependent process (Lai et al, 1999). ZIKV's envelop protein E binds to a C-type lectin receptor (Hamel et al, 2015) (e.g., DC-SIGN—dendritic cell-specific intercellular adhesion molecule-3-grabbing non-integrin) at the host cell surface to enable the viral entry process (Agrelli et al, 2019). Remarkably, ZIKV fails to mature in calcium pump SPCA1-deficient cells, preventing viral spread. SPCA1 is predominantly localized to the Golgi apparatus. This indicates that an appropriate $Ca^{2+}$ concentration in the Golgi apparatus is critical for late stages of the viral life cycle (Hoffmann et al, 2017).

The transient receptor potential (TRP) proteins represent a superfamily of cation channels that are permeable to $Ca^{2+}$ and have been involved in diverse physiological processes in the brain, as well as in the pathogenesis of many neurological diseases (Chen et al, 2020). TRP channels can be stimulated by a variety of mechanisms such as ligand binding, voltage, physical, and chemical stimuli. The studies with genetic deletion of TRPC family members have demonstrated that TRPC channels, particularly TRPC1/4 and TRPC5 channels, play a critical role in chemically induced acute seizures and neuronal cell death (Phelan et al, 2013). It remains unknown whether the host TRPC expression or TRPC function is altered during ZIKV infection, and whether this contributes to ZIKV-associated seizure or death.

In this study, we found that ZIKV infection resulted in an increased host TRPC4 protein expression via the CaMKII–CREB pathway. Inhibitors of the TRPC4 channel and CaMKII blocked the propagation of ZIKV in human brain organoids and improved the symptoms of epilepsy and survival in a neonatal mouse model of ZIKV infection.

## Results

### The expression of TRPC4 was upregulated in the ZIKV-infected cells and mouse brain

$Ca^{2+}$ permeable TRPC4 channels are widely expressed in the brain, and TRPC4 activation may represent an important aspect of ZIKV pathogenesis. We first noticed that host TRPC4 protein levels were increased by ZIKV (SMGC-1 strain) infection in BHK (Baby Hamster Syrian Kidney) cell monolayers which were inoculated with ZIKV at a multiplicity of infection (MOI) of 0.001 or 0.01. The efficiency of ZIKV infection was monitored by assessing the ZIKV-non-structural protein 1 (NS1) protein expression. At 72 h post infection (hpi), TRPC4 protein levels in the ZIKV-infected cells were assessed by using immunofluorescence staining and western blots. The immunofluorescence staining data indicated that the TRPC4 protein expression in BHK cells was elevated proportionally to increasing ZIKV viral load (Fig. 1A,D). Analysis of the data revealed that there was a strong positive correlation between the TRPC4 and ZIKV E-protein levels (Fig. 1B, $r = 0.83$). To establish that the observed effect is not restricted to BHK cells, we next infected human astrocytoma U87 cells with ZIKV. We found that TRPC4 protein levels were also proportionally greater in the protein lysates isolated from U87 cells infected with higher ZIKV MOI (Fig. 1C). Quantification of the western blot data obtained using BHK cell lysates yielded similar results. ZIKV-infected BHK cells exhibited $1.58 \pm 0.15$ greater TRPC4 protein levels compared to mock-infected BHK cells ($n = 6$, Fig. 1D). Conversely, U87 cells

heterologously overexpressing the TRPC4 channel produced elevated levels of the ZIKV-NS1 protein after being infected with ZIKV (Fig. 1E).

We subsequently compared the RNA expression levels of TRPC4 and ZIKV in BHK cells at 48 h and 72 h post infection. The relative TRPC4 RNA levels time-dependently increased, reaching $1.4 \pm 0.1$ at 48 h and $3.1 \pm 0.2$ at 72 h, which positively correlated with the extent of ZIKV propagation (Fig. 1F, left). The relative ZIKV RNA levels were $408.8 \pm 115.5$ at 48 h and $1470.2 \pm 289.6$ at 72 h (Fig. 1F, right). Conversely, no significant alterations were observed in the expression of TRPC1, TRPC3, TRPC5, or TRPC6 upon ZIKV infection (Appendix Table S1).

We then set out to determine whether our in vitro findings can be further confirmed in an in vivo mouse model. Since ZIKV is quickly cleared in ZIKV-inoculated wild-type adult C57BL/6 mice, we used interferon receptor-deficient adult A129 mice ($Ifnar^{-/-}$ mice) and 1-day-old suckling ICR mice to detect whether ZIKV infection affects the TRPC4 level in the mouse brain. Compared to brains of mock-infected mice, the brains of ZIKV-infected adult mice showed increased immunostaining of glial fibrillary acidic protein (GFAP) in the proximity of lesions and in the CA1 and dentate gyrus hippocampal regions (Fig. EV1A) at 12 days post infection (dpi). The expression of NeuN (neuronal nuclear protein) remained unaltered in A129 adult mouse brains following ZIKV inoculation (Fig. EV1B), whereas a decrease was observed in suckling ICR mouse brains (Fig. EV1C). Figure EV1B,C shows that the immunostaining intensity for the TRPC4 protein (green) was increased in the ZIKV-inoculated A129 and sucking ICR mouse brains compared to the mock group at 12 dpi. The immunofluorescence analysis of the adult mouse brain images demonstrated a low infection rate of mature neurons (red) by ZIKV, which is consistent with the findings reported in the literature (Li et al, 2016). Notably, the TRPC4 protein exhibited a significantly elevated expression level in the specific region where ZIKV was detected in the adult mouse brain, with a remarkably high correlation coefficient of 0.91 (Fig. EV1B, right inset). The correlation between TRPC4 and ZIKV RNA levels in the brains from ZIKV-infected A129 mice also had a positive correlation coefficient of $r = 0.86$ (Fig. 1G). Immunoblots were next used to determine the TRPC4 and ZIKV protein levels in the brain of mock and ZIKV-inoculated A129 mice. However, the correlation between TRPC4 and ZIKV-NS1 proteins was not as robust as that observed for RNA levels. This discrepancy may be attributed to our utilization of protein lysates derived from whole brain tissue without subdivision into specific cell types. Nevertheless, Fig. 1H demonstrates an elevation in TRPC4 protein levels within the ZIKV-NS1 positive areas. Taken together, these results suggested that ZIKV promotes host cell TRPC4 expression in vitro and in vivo.

### Downregulation of TRPC4 expression with RNAi or inhibition of TRPC4 channels using HC-070 led to an increased ZIKV-infected cell survival and a reduced ZIKV protein production

To further validate the role of host TRPC4 in ZIKV infection, an shRNA targeting TRPC4 (shRNA-TRPC4) was used to knock-down the expression of TRPC4 in BHK cells (Fig. 2A,B), while utilizing scrambled RNA (scRNA) as a control. Immunoblotting

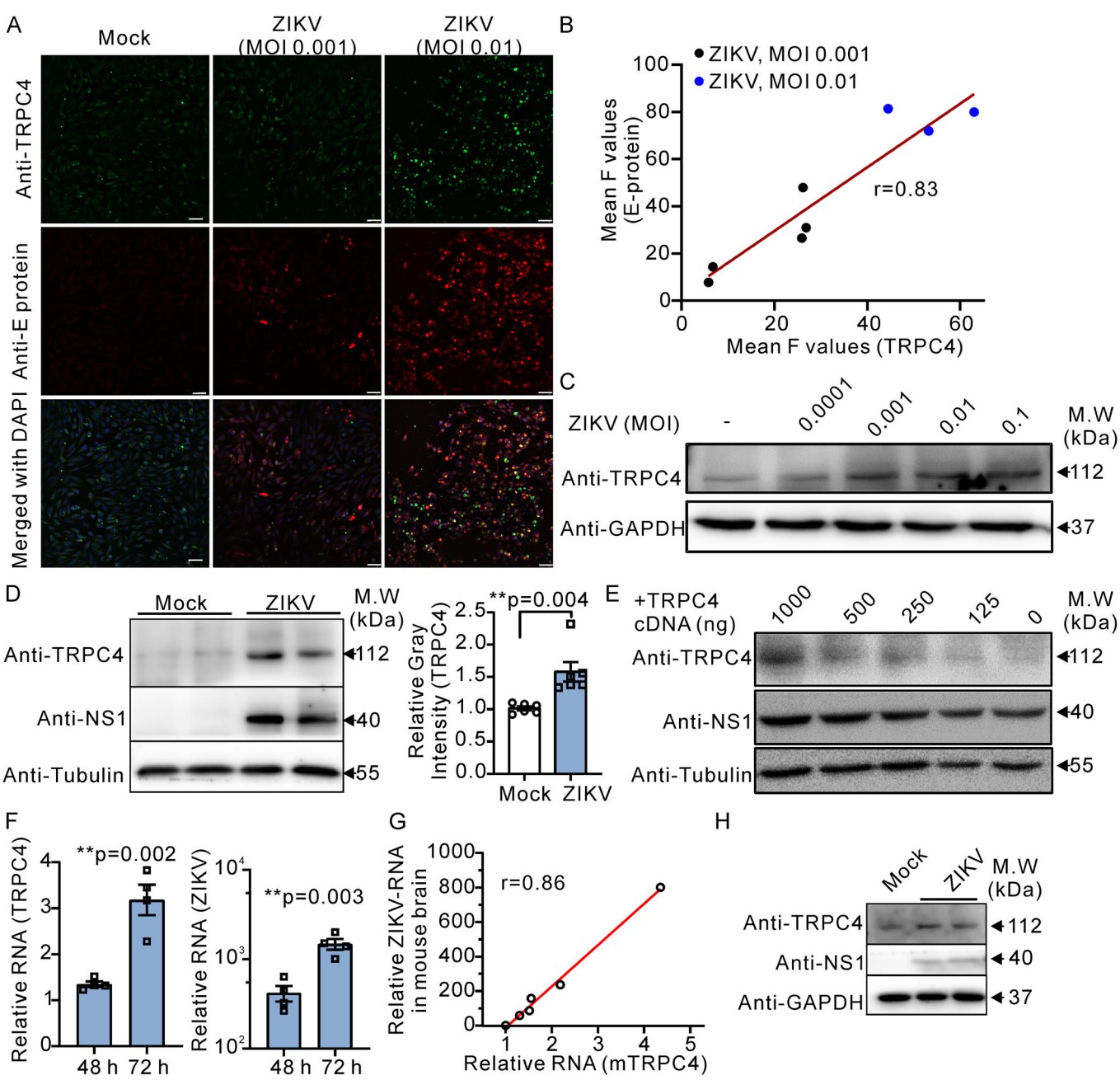

**Figure 1. TRPC4 expression is elevated in host cells and mouse brains upon ZIKV infection.**

(A) Representative immunofluorescence images of ZIKV-infected wild-type BHK cells taken at 72 hpi. The scale bar represents 50 μm. (B) Pearson correlation analysis. The correlation plot shows strong correlations between TRPC4 protein expression and ZIKV E-protein production ($r = 0.83$). The fluorescence intensity (F) was quantified using ImageJ software. (C) Representative immunoblot images show relatively elevated TRPC4 protein amounts in U87 cells infected with higher MOI of ZIKV at 72 hpi. (D) Representative western blot images show TRPC4, NS1, and tubulin protein expression in ZIKV-infected or Mock BHK cells (left). Right, a comparison of the relative gray intensity for TRPC4 in western blots depicted on the left ($n = 6$ biological replicates), with tubulin staining being used as a loading control. (E) Western blot analysis of the ZIKV viral NS1 protein expression in U87 cells overexpressing variable amounts of the TRPC4 cDNA and subjected to a 72-h challenge with the ZIKV virus. (F) The shown plots compare the relative RNA levels of TRPC4 (left) and ZIKV (right, $n = 4$ biological replicates) in BHK cells infected with ZIKV at 48 hpi and 72 hpi. qRT-PCRs were performed to amplify the TRPC4 and ZIKV (the linker region of membrane protein and envelope protein) RNA. (G) The correlation analysis. There is a strong correlation between mRNA levels of TRPC4 and ZIKV (NS5) in the ZIKV-infected mouse brains ($r = 0.86$). (H) Representative immunoblot images are shown. Mock- or ZIKV-infected mouse brains were collected to perform the western blot analysis. Increased TRPC4 protein amounts were detected in ZIKV-infected mouse brains compared to Mock (control)-infected brains. The unpaired $T$ test (two-tailed) was used to determine if there was a significant difference between two groups, and the correlation coefficient was obtained through linear regression analysis. Data information: In (D, F), data are presented as mean ± SEM, $**P \leq 0.01$. Source data are available online for this figure.

data confirmed that the ZIKV-NS1 protein expression was significantly decreased in ZIKV-infected shRNA-TRPC4-BHK cells, expressing reduced levels of the TRPC4 protein ($0.6 \pm 0.1$, Fig. 2C), compared to control ZIKV-infected scRNA-BHK cells. Notably, we observed a significantly increased cell survival in the ZIKV-infected cells with downregulated TRPC4 ($9.6 \pm 2.0\%$ *vs* $69.2 \pm 5.1\%$; Fig. 2D). In addition, we also investigated whether the downregulation of TRPC4 leads to a reduction in ZIKV-NS1 protein production in ZIKV-infected HT22 cells (immortalized mouse embryonic hippocampal neuronal cell line). As depicted in Fig. EV1D, siRNA-mediated knockdown of TRPC4 significantly attenuated the production of NS1 ($1.01 \pm 0.03$ *vs* $0.55 \pm 0.02$, $P = 0.002$).

In addition, we investigated the potential involvement of other TRP channels in ZIKV infection by evaluating the antiviral activity of the channel modulators (Fig. 3A; Table 1). BHK cell monolayers were treated for 1 h with each modulator, vehicle, or NITD008 (Deng et al, 2016) as the positive control, and then inoculated with ZIKV at an MOI of 0.001 or mock. At 7 dpi, cell viability was assessed using the CellTiter-Glo Luminescent kit. We found that HC-070, a potent TRPC4/5 inhibitor, significantly increased BHK cell survival (from $16.4 \pm 0.9\%$ to $88.3 \pm 1.9\%$, $P < 0.001$) following ZIKV infection with an $IC_{50}$ value of $4.5 \pm 3.0\ \mu M$ ($n = 3$). HC-070 exhibited a similar potency against DENV infection in the same BHK cell infection model (Table 1, $IC_{50} = 6.6 \pm 1.3$, $n = 3$). ML204, a more specific TRPC4 inhibitor, also showed anti-ZIKV activity but with a higher $IC_{50}$ value of $58.0 \pm 4.5\ \mu M$ ($n = 3$). However, AC1903, a specific TRPC5 inhibitor (Zhou et al, 2017), and Pyr3, an antagonist for TRPC3/6 (Glasnov et al, 2009), did not exhibit protective properties in our cell model for ZIKV infection. The cell viability was not improved by either AC1903 ($5.8 \pm 3.6\%$, $n = 3$) or Pyr3 ($15.8 \pm 0.9\%$, $n = 3$). The modulators exhibited negligible cytotoxic effects on the cells in the control mock group (Fig. EV1E). The anti-ZIKV activity of HC-070 was also tested in Huh7, Vero, H4, U87, and A127 cell lines (Fig. 3B). The treatment with HC-070 significantly reduced the levels of ZIKV RNA copies in BHK, H4, and U87 cells. The HC-070 treated group, but not the AC1903 treated group, exhibited a reduction in the production of viral nonstructural protein 1 (NS1) as assessed by western blots (Fig. 3C, $0.4 \pm 0.1$, $n = 4$). In addition, the reduced ZIKV infection in the presence of HC-070 was associated with significant downregulation of TRPC4 expression (Fig. 3D). Moreover, Fig. 3E,F shows that HC-070 treatment significantly inhibited expression of ZIKV E proteins (relative red fluorescence intensity, $0.3 \pm 0.1$, $n = 5$) compared to the vehicle control ($1.0 \pm 0.1$) or the AC1903 ($1.2 \pm 0.1$) treated group. Furthermore, HC-070 concentration-dependently inhibited the production of ZIKA RNA and ZIKV-NS1 proteins (Fig. 3G,H). Notably, the suppressive effect of HC-070 on viral protein expression was already observed at an inhibitor concentration of ~1 $\mu M$, which aligns with its demonstrated protective effects on infected cells as depicted in Fig. 3A.

## ZIKV increases TRPC4 expression via the CaMKII–CREB pathway

Given the predominant distribution of TRPC4 proteins in the plasma membrane, it is plausible that this channel may function as a receptor for the viral entry of ZIKV's E surface protein mediating the viral entry. To examine this hypothesis, we performed co-

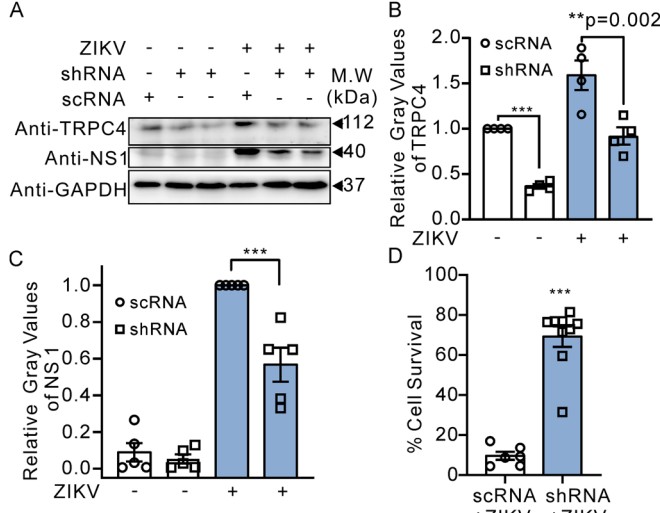

**Figure 2. Downregulation of TRPC4 resulted in a reduction of ZIKV infection in BHK cells.**

(A) Representative immunoblot images are shown. The treatments are indicated above the immunoblots. (B, C) The shown plots demonstrate that ZIKV-infected shRNA-TRPC4-BHK cells exhibited decreased levels of TRPC4 and ZIKV-NS1 proteins compared to ZIKV-infected scRNA-BHK cells. TRPC4 protein expression was also decreased in mock-infected shRNA-TRPC4-BHK cells compared to mock-infected scRNA-BHK cells. Western blots were used to compare TRPC4 ($n = 4$ biological replicates) and NS1 ($n = 5$ biological replicates) protein expression in the indicated groups. (D) The percentage of cell survival was increased to 68% in the TRPC4-shRNA group ($n = 9$ biological replicates) compared to the scRNA control group (9.6%). The unpaired *T* test (two-tailed) and one-way ANOVA followed by the Tukey test as the post hoc were employed to determine if there was a significant difference between two groups or among multiple groups, respectively. GAPDH staining was used as a loading control. Data information: In (B–D), data are presented as mean ± SEM, **$P \leq 0.01$, ***$P \leq 0.0001$. Source data are available online for this figure.

immunostaining of TRPC4 and ZIKV-E proteins. However, the co-localization analyses indicated that there was little or no interaction between the TRPC4 protein and viral E protein in cells (Fig. EV2A,B, Pearson coefficient, 0.46). Furthermore, BHK cells were transfected with TRPC-flag plasmids and subsequently infected with ZIKV to investigate the interaction between TRPCs and ZIKV-E protein using Co-IP analysis. As depicted in Fig. EV2C, no discernible interaction was observed. Therefore, we concluded that the TRPC4 protein does not serve as a ZIKV receptor.

The above results indicated that the host TRPC4 protein expression was elevated after ZIKV infection and that the increased TRPC4 protein levels were necessary for ZIKV spread. It was reported that TRPC4 protein abundance in the cell plasma membrane depends on the cytosolic $Ca^{2+}$ level and is regulated via the CaMKII–CREB (cAMP-response element binding protein) pathway (Morales et al, 2007). Both calcium influx and calcium release from its intracellular stores may induce an increase in TRPC4 mRNA and protein levels (Morales et al, 2007). Remarkably, some viral infections are accompanied with an increase of the host cytosolic $Ca^{2+}$ level (Donate-Macian et al, 2018; Fujioka et al, 2018).

Therefore, we next set out to test whether ZIKV infection is associated with changes in intracellular $Ca^{2+}$ levels. In these

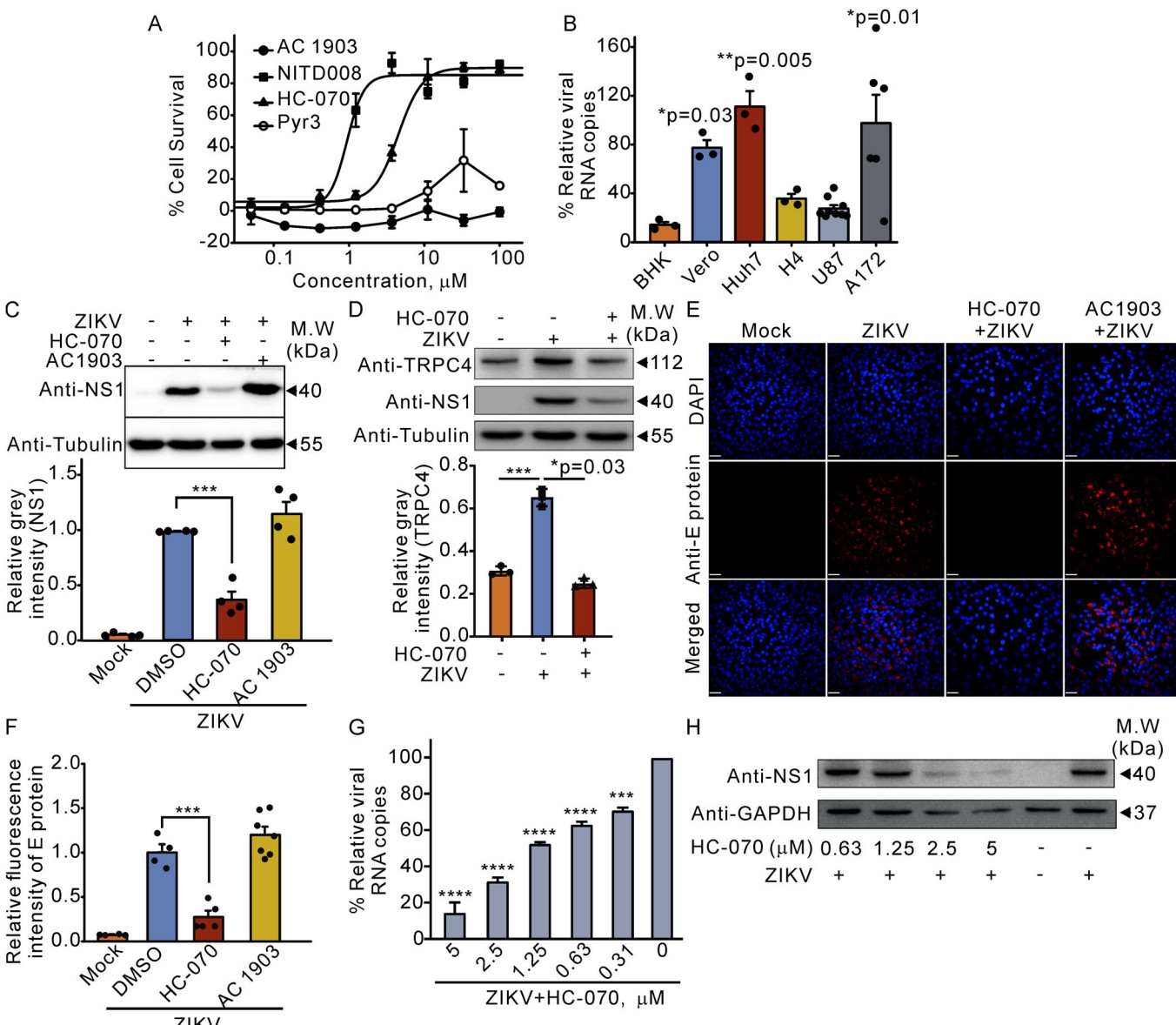

**Figure 3.  Treatment with a TRPC4 inhibitor, HC-070, reduces the production of the ZIKV protein in BHK cells.**

(A) Cell viability following ZIKV infection in the presence of specific channel modulators ($n = 3$ biological replicates). (B) The plot shows the results of qRT-PCR quantification of relative ZIKV RNA copies in the mRNA isolated from various cell lines infected with ZIKV in the presence of HC-070 ($n = 3$–9 biological replicates). (C, D) Representative immunoblots and the quantification results. BHK cells were seeded on six-well plates and pretreated with HC-070 (10 μM), AC1903 (10 μM, a specific TRPC5 inhibitor), or the vehicle control for 1 h and then infected with ZIKV (SMGC-1, MOI = 0.001). Cell lysates were collected at 72 hpi. The western blot analysis was used to quantify the ZIKV-NS1 ($n = 4$ biological replicates) and TRPC4 ($n = 3$ biological replicates) protein levels. Bands' intensities were quantified using ImageJ software. Tubulin staining was used as a loading control. The plot in the lower panel of C shows that HC-070 but not AC1903 decreased the ZIKV-NS1 protein level in ZIKV-infected cells. The plot in the lower panel of D shows that HC-070 decreased the TRPC4 protein level in ZIKV-infected cells. (E, F) Cells were inoculated in 96-well plates treated as above. Seventy-two hours later, cells were stained with the anti-ZIKV-E antibody. Representative images of BHK immunostaining for the ZIKV E protein (red) and summary data of mean F intensity ($n = 4$-7 biological replicates). The scale bar represents 50 μm. (G, H) The production of ZIKV RNA ($n = 3$ biological replicates) and ZIKV-NS1 proteins was attenuated in BHK cells treated with a series of concentrations of HC-070. The Dunnett's test as the post hoc following one-way ANOVA on ranks (B) or The Student–Newman–Keuls Method as the post hoc following one-way ANOVA (C–G) was employed to determine if there was a significant difference among multiple groups. Data information: In (A–D, F, G), data are presented as mean ± SEM, *$P \leq 0.05$, **$P \leq 0.0001$, ****$P \leq 0.0001$. Source data are available online for this figure.

**Table 1. Comparison of anti-flavivirus activity of various TRP channel modulators that are commonly used.**

| | | IC$_{50}$ (µM) | | | |
|---|---|---|---|---|---|
| Modulators | Channels | Anti-ZIKV | Anti-DENV | Anti-YFV-17D | Anti-JEV-SA14 |
| [a]NITD008 | Posi positive control | 1.24 ± 0.67 | 5.46 ± 0.09 | 1.62 ± 0.03 | 3.94 ± 0.06 |
| GSK1702934A | Agonist for TRPC3/C6 | >200 | 22.39 ± 7.23 | >200 | >200 |
| Capsaicin | Agonist for TRPV1 | >200 | >200 | >200 | >200 |
| Trans-Cinnamaldehyde | Agonist for TRPA1 | 56.05 ± 4.81 | >200 | >200 | >200 |
| Icillin | Agonist for TRPM8 | >200 | >200 | >200 | >200 |
| Pyr3 | Antagonist for TRPC3/C6 | >200 | >200 | >200 | >200 |
| **ML204** | **Antagonist for TRPC4/C5** | **58.01 ± 4.45** | **>200** | >200 | >200 |
| **HC-070** | **Antagonist for TRPC4/C5** | **4.54 ± 3.00** | **6.59 ± 1.29** | >200 | >200 |
| Pico145 | Antagonist for TRPC4/C5 | >200 | >200 | >200 | >200 |
| Galangin | Antagonist for TRPC4/C5 | >200 | >200 | >200 | >200 |
| **AC1903** | **Antagonist for TRPC5** | **>200** | **>200** | >200 | >200 |
| RQ-00203078 | Antagonist for TRPM8 | >200 | >200 | >200 | >200 |
| Capsazepine | Antagonist for TRPV1 | >200 | >200 | >200 | >200 |
| RN-1734 | Antagonist for TRPV4 | 11.61 ± 2.12 | >200 | >200 | >200 |
| HC-067047 | Antagonist for TRPV4 | 5.10 ± 1.15 | >200 | >200 | >200 |
| Imperatorin | Antagonist for TRPV1 | >200 | >200 | >200 | >200 |
| 2-Hydroxyestradiol | Activate TRPA1 | >200 | >200 | >200 | >200 |
| LaCl$_3$ | Activate TRPC4/C5 | 5.57 ± 1.38 | 30.85 ± 5.66 | 2.48 ± 2.33 | 67.37 ± 9.82 |
| GdNO$_3$ | Activate TRPC4/C5 | 13.04 ± 4.52 | >200 | >200 | >200 |
| 2-APB | TRP channel modulator | 58.56 ± 13.26 | >200 | >200 | >200 |

*ZIKV* Zika virus, *DENV* dengue virus, *YFV* yellow fever virus, *JEV* Japanese encephalitis virus.
[a]NITD008 was used as a positive control.
The bold values represent TRPC4/5 antagonists with anti-ZIKV activity, as well as a TRPC5 antagonist lacking anti-ZIKV activity.

experiments, we resorted to using the cells expressing the red fluorescent genetically encoded Ca$^{2+}$ indicators (R-GECO). Indeed, we observed cytosolic Ca$^{2+}$ concentration increases in the ZIKV-infected R-GECO-expressing cells (Fig. EV2D) during the early stage of viral infection. To determine whether extracellular Ca$^{2+}$ plays a role in cytosolic Ca$^{2+}$ level rises after ZIKV infection, we next treated BHK cells with medium supplemented with 1 mM EGTA to chelate extracellular Ca$^{2+}$ ions. Remarkably, EGTA treatment significantly suppressed viral E-protein production while enhancing cell resilience against ZIKV infection (Fig. 4A,B). This suggests that the viral life cycle relies on the influx of Ca$^{2+}$ from outside via Ca$^{2+}$ permeable channels. Figure 4A also shows that the survival of ZIKV-infected cells increased in the presence of HC-070 (10 µM), implicating TRPC4 activity in promoting viral replication that leads to cell death.

It was reported that ZIKV-NS3 protein, an RNA helicase, could stimulate the phosphorylation of CaMKIIα in host cells (Sun et al, 2020). Therefore, we reasoned that inhibition of CaMKII to downregulate the TRPC4 protein level might also inhibit ZIKV propagation. To test this hypothesis, we treated BHK cells with KN-93, an inhibitor of CaMKII, before infecting them with ZIKV. We found that BHK cell viability and survival following ZIKV infection increased from 21.0 ± 4.0% to 98.6 ± 2.9% (Fig. 4A) in the presence of KN-93 (10 µM). The immunostaining and western blot analysis revealed that, like HC-070, KN-93 (10 µM) effectively inhibited ZIKV-E and ZIKV-NS1 protein production (Fig. 4B,C). We next

collected the supernatant of cell cultures to determine the effect of HC-070 or KN-93 treatment on the number of released infectious viral particles. We used the viral plaque-forming unit (PFU) assay during these experiments. Not surprisingly, in line with the above results, we found that both HC-070 (10 µM) and KN-93 (10 µM) effectively decreased the PFU of ZIKV (Fig. 4D). Treatment with KN-93 (10 µM) also led to a reduction in the protein level of TRPC4 (Fig. 4E). Furthermore, co-expression of ZIKV-NS3 and TRPC4 cDNAs in R-GECO-expressing cells led to an elevated background cytosolic Ca$^{2+}$ concentration (Fig. EV2G) due to the augmented expression rate of TRPC4 protein (Fig. EV2E,F). This sustained increase was effectively suppressed by HC-070, as demonstrated in Fig. EV2G. The interaction between CaMKII and overexpressed ZIKV-NS3 was also observed (Fig. 4F), indicating that the NS3 protein may modulate CaMKII activity, which is consistent with previous research findings (Sun et al, 2020).

We next performed experiments to determine the involvement of CREB in regulating TRPC4 expression in ZIKV-infected cells. Immunostaining for pCREB of the mouse brain revealed significantly increased levels of pCREB in ZIKV-infected mice (Figs. 4G and EV3). In addition, using western blot analysis, we found that ZIKV-infected BHK cells treated with KN-93 (10 µM) or EGTA (1 mM) exhibited significantly reduced levels of phospho-CREB (pCREB, Ser133; Fig. 4H, 0.68 ± 0.08 and 0.31 ± 0.11, respectively) compared to vehicle control (DMSO, 1.28 ± 0.16)-treated cells. However, HC-070 treatment did not affect pCREB

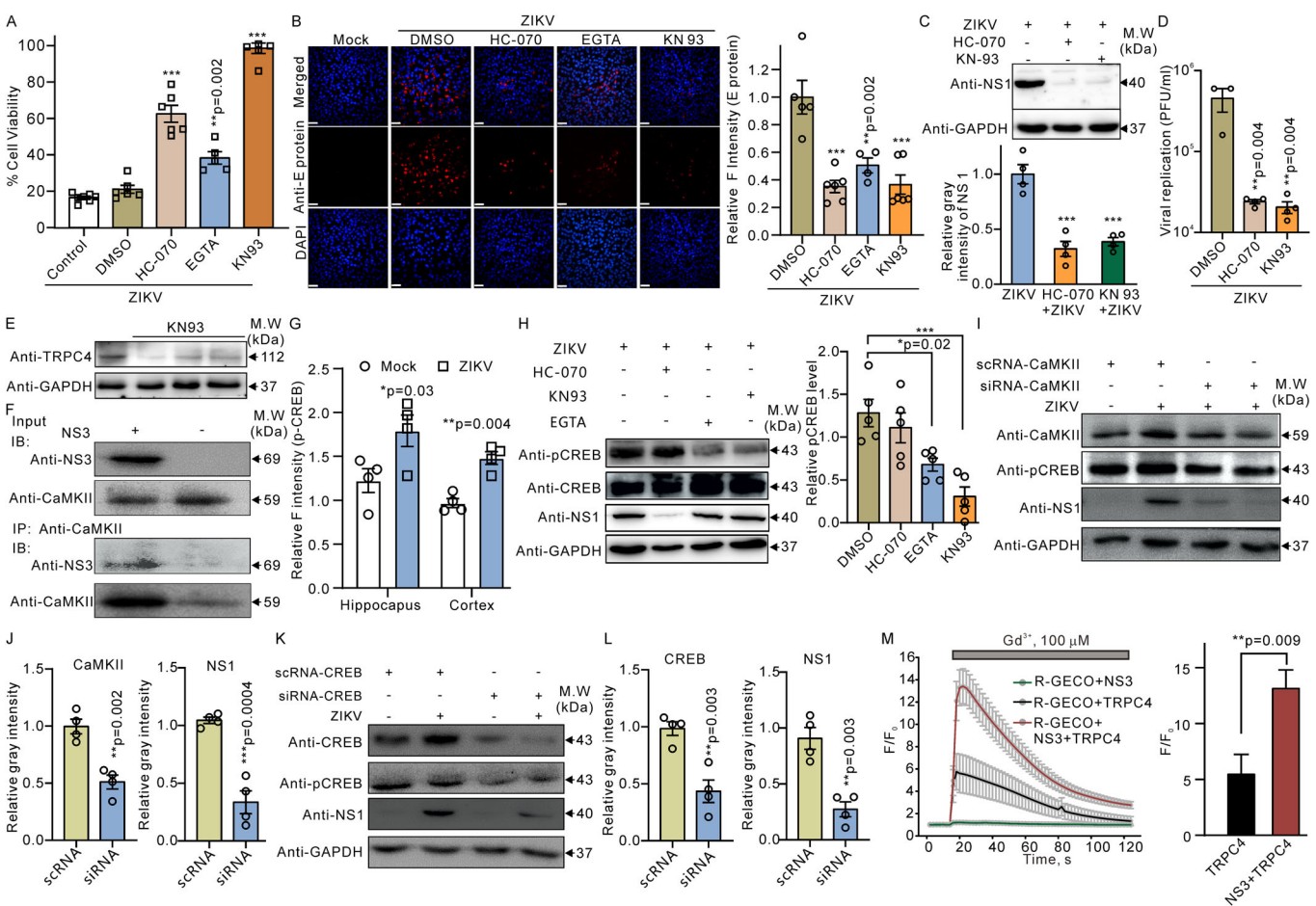

**Figure 4. Downregulation of CaMKII decreases the production of ZIKV protein in host cells.**

(A) Survival rates of ZIKV-infected BHK cells were increased in the HC-070 (10 µM), EGTA (1 mM, the extracellular calcium chelating agent), and KN-93 (10 µM, CaMKII inhibitor) treatment groups ($n = 5$–7 biological replicates). (B) Representative images (left) and summary data (right) of ZIKV- or mock-infected BHK cells immunoassayed for the ZIKV E protein (red). The cells were pretreated with DMSO, HC-070 (10 µM), EGTA (1 mM), or KN-93 (10 µM). The fluorescence intensity (F) was quantified using ImageJ software, and the obtained data were compared in the right panel ($n = 4$–6 biological replicates). The scale bar represents 50 µm. (C) Western blots were performed to detect and quantify the production of ZIKV-NS1 proteins (NS1) in KN-93 (10 µM) or HC-070 (10 µM) pretreated BHK cells infected with ZIKV ($n = 4$ biological replicates). (D) Viral replication was quantified by counting infectious viral particles isolated from the supernatants of BHK cells infected with ZIKV and treated with either DMSO (vehicle control), HC-070 (10 µM), or KN-93 (10 µM). Infectious viral particles were detected using the viral plaque-forming unit assay ($n = 4$ biological replicates). (E) Representative immunoblot images showing a decrease of TRPC4 protein levels in KN-93-treated BHK cells without ZIKV infection. (F) Co-immunoprecipitation (Co-IP) assays are shown. Whole-cell extracts from ZIKV-NS3 overexpressed BHK cells were subjected to immunoprecipitation (IP) using either anti-NS3 or anti-CaMKII antibody. The co-immunoprecipitated proteins were subsequently detected by western blotting using specific antibodies against CaMKII or NS3. (G) The graph shows that ZIKV-infected neonatal mouse brains exhibit an enhanced immunofluorescence staining of pCREB ($n = 4$ mice) compared to mock-infected mice. (H) Western blot analysis was performed to detect (left) and quantify (right) the phosphorylation level of CREB (pCREB) relative to total CREB in KN-93 (10 µM) or HC-070 (10 µM) pretreated BHK cells infected with ZIKV ($n = 5$ biological replicates). (I–L) Representative immunoblot images show the efficacy of siRNA-mediated knockdown of the ZIKV-NS1 protein production in ZIKV-infected BHK cells ($n = 4$ biological replicates). Prior to ZIKV challenge (MOI 0.01), cells were transfected with scRNAs or siRNAs targeting CaMKII (I, J) or CREB (K, L) for a duration of 48 h ($n = 4$ biological replicates). (M) Shown are the changes in normalized R-GECO fluorescence ($F/F_o$) induced by $Gd^{3+}$ (100 µM) in TRPC4 (black line), NS3 (green line), or TRPC4 + NS3 (red line)-expressing cells. Intracellular $Ca^{2+}$ levels were monitored using the R-GECO biosensor. Cells were transfected with the indicated cDNAs. The horizontal bar shows the times when $Gd^{3+}$ was added to the wells with cells. The right panel displays a comparison of normalized R-GECO fluorescence increases in each tested group ($n = 9$–12, 3–4 wells were for each experiment, with three biological replicates). The unpaired T test (two-tailed) and one-way ANOVA followed by the Dunnett's test as the post hoc were employed to determine if there was a significant difference between two groups or among multiple groups, respectively. Data information: In (A–D, G, H, J, L, M), data are presented as mean ± SEM, *$P \le 0.05$, **$P \le 0.01$, ***$P \le 0.001$. Source data are available online for this figure.

levels. Our western blot data confirmed that HC-070 (10 µM) attenuated the production of ZIKV-NS1 protein (Fig. 4C).

To further validate the role of the CaMKII-pCREB during ZIKV infection, the siRNA approach was employed to knock down either the CaMKII or CREB protein in BHK cells. The results depicted in Fig. 4I–L demonstrate a significant reduction of NS1 viral protein

levels following ZIKV infection in CaMKII knockdown BHK cells (1.05 ± 0.03 vs 0.34 ± 0.10, Fig. 4I,J). In addition, we found that a decrease of CREB levels was associated with reduced NS1 production (0.91 ± 0.10 vs 0.27 ± 0.07, Fig. 4K,L). The findings were replicated in HT22 cells, as depicted in Fig. EV3B,C. Thus, ZIKV-NS3-dependent activation of $Ca^{2+}$/CaMKII and phosphorylation of

CREB may account for the increase in TRPC4 transcription in response to ZIKV infection.

We next conducted live-cell calcium imaging experiments to investigate the impact of acute KN-93 (10 µM) administration on TRPC4 function. As depicted in Fig. EV3D, KN-93 did not exert any acute influence on TRPC4-mediated calcium influx. Thus, the data indicate that there is no direct regulatory role of CaMKII, via phosphorylation or protein-protein interaction, in relation to TRPC4.

We next assessed whether ZIKV-NS3 affects the function of the TRPC4 channel. In these experiments, we used live-cell calcium imaging in BHK cells. TRPC4 channels can be activated either by signaling molecules downstream of G protein-coupled receptors (GPCRs) or by trivalent cations like $Gd^{3+}$ and $La^{3+}$. Initially, we used $Gd^{3+}$ (100 µM) to activate TRPC4 (Fig. 4M). As depicted in Fig. 4M, the presence of ZIKV-NS3 protein significantly augmented the calcium influx mediated by TRPC4 in R-GECO-BHK cells. We later confirmed the findings in BHK cells co-expressing R-GECO and H1 histamine receptor (H1R), in addition to either ZIKV-NS3, TRPC4, or ZIKV-NS3 + TRPC4. Consistently, we found that histamine (10 µM) induced larger intracellular $Ca^{2+}$ increases in ZIKV-NS3 + TRPC4 + R-GECO + H1R-expressing cells compared to TRPC4 + R-GECO + H1R-expressing BHK cells (Fig. EV3E).

## HC-070 inhibits ZIKV replication

The literature provides evidence supporting a vital role of the organellar $Ca^{2+}$ dynamics in regulating virus entry, replication, and severity of the infection (Saurav et al, 2021). To further investigate the mechanisms underlying the antiviral activity of HC-070, we initially conducted a time-of-drug-addition assay (Yan et al, 2022) to explore which specific stage of the ZIKV life cycle was affected by HC-070. BHK cells infected with ZIKV were treated with HC-070 at different stages of the infection (Fig. 5A). The positive control drug NITD008, a nucleotide analog known to inhibit the stage of viral intracellular replication (Yan et al, 2022), prevented viral RNA production in stages II–IV (Fig. 5B), corresponding to the replication stage of the ZIKV life cycle. HC-070 also significantly suppressed ZIKV RNA production only in stages II–IV (Fig. 5B), although there was a relatively small inhibition of viral replication by HC-070 in stage IV. These data indicate that calcium influx through TRPC4 may be essential for the whole stage of replication and highlight the vital role of the host TRPC4 channel for viral propagation. This is consistent with a general assumption that the elevation of intracellular $Ca^{2+}$ is a crucial factor for the replication of the viruses (Donate-Macian et al, 2018).

We then performed a temperature-dependent infectivity inhibition assay to assess the impact of HC-070 and KN-93 on ZIKV viral particle virulence. In brief, we incubated viral particles with either HC-070, KN-93, or EGCG (epigallocatechin gallate, the positive control) at specified temperatures and then performed a plaque assay to measure virus infectivity. As depicted in Fig. 5C, neither HC-070 nor KN-93 exhibited a significant effect on ZIKV virulence.

It was reported that the host RNA DEAD-box helicase (DDX3X) plays a crucial role in ZIKV replication (Nelson et al, 2021; Riva et al, 2020). Therefore, we next tested whether TRPC4-mediated calcium influx affects the nuclear localization of DDX3X. We found that the ZIKV infection resulted in the accumulation of DDX3X in the nucleus (identified by the co-localization of DDX3X and DAPI

nuclear staining) in BHK cells (Fig. 5D). The right panel in Fig. 5D shows the average E-protein density and the ratio of nuclear/total DDX3X intensity observed under all experimental conditions tested. Treatment with HC-070 (10 µM) or EGTA (1 mM) significantly reduced the nuclear/total DDX3X intensity ratio following ZIKV infection, with values of $0.40 \pm 0.07$ and $0.58 \pm 0.11$, respectively. Furthermore, nuclear proteins were extracted from cells overexpressing TRPC4 or control cells, and western blot analysis was performed to determine the abundance of DDX3X in the nuclear fraction. Significantly elevated levels of DDX3X were observed in the nuclei of ZIKV-infected TRPC4-overexpressing cells compared to control cells (Fig. 5E). These data provide evidence for the potential involvement of TRPC4 in regulating ZIKV replication likely via modulating the host DDX3X nuclear localization.

## HC-070 and KN-93 inhibit ZIKV spread in human brain organoid

ZIKV is capable of infecting human iPSC (induced pluripotent stem cell)-derived brain organoids, and this approach has been used to investigate the mechanisms of ZIKV entry (Fan et al, 2021). This model allows investigation of pathological processes associated with ZIKV infection affecting the nervous system. Therefore, we next employed the human iPSC-derived brain organoid model to test whether HC-070 or KN-93 exhibits any protective effects against ZIKV infection. ZIKV (SMGC-1 strain, $1.2 \times 10^6$ PFU per organoid) readily infected 30-day brain organoids which were engineered to bear a luciferase reporter system. Such brain organoids would release large amounts of luciferase into the supernatant due to the organoid rupture caused by ZIKV infection. After 2 days, the supernatants and organoids were collected to quantify organoid survival and ZIKV RNA levels. The Nano-Glo luciferase assay was used to assess the survival of brain organoids after ZIKV infection in the HC-070, KN-93, and vehicle-treated groups (Fig. 6A). The luminescence was significantly smaller in the HC-070 and KN-93 group supernatants compared to the vehicle group (Fig. 6B), which indicated a reduced organoid rupture. The viral RNA copy number was also decreased in the organoids and supernatant samples of HC-070 and KN-93-treated groups compared to the control (Fig. 6C,D), indicating a reduced viral replication. Thus, both HC-070 and KN-93 exhibited potent protective effects against ZIKV infection. These data further support the hypothesis that elevated host TRPC4 expression induced by ZIKV infection-driven CaMKII activity is necessary for virus replication.

## HC-070 and KN-93 protect ICR mice from lethal ZIKV challenges

We next tested whether TRPC4 activity or TRPC4 expression regulation may contribute to ZIKV-related deaths and/or seizures. One-day-old suckling ICR mice were first intraperitoneally infected with $2 \times 10^4$ PFU of ZIKV. These ZIKV-infected mice were then intraperitoneally treated with HC-070 (0.3 and 1.0 mg/kg/day), KN-93 (3 mg/kg/day) or DMSO (vehicle control) at 2 dpi and then daily for 10 consecutive days. Body weights and survival of the ZIKV-infected mice were monitored until 21 dpi. Body weights were not significantly increased in HC-070 and KN-93 treated

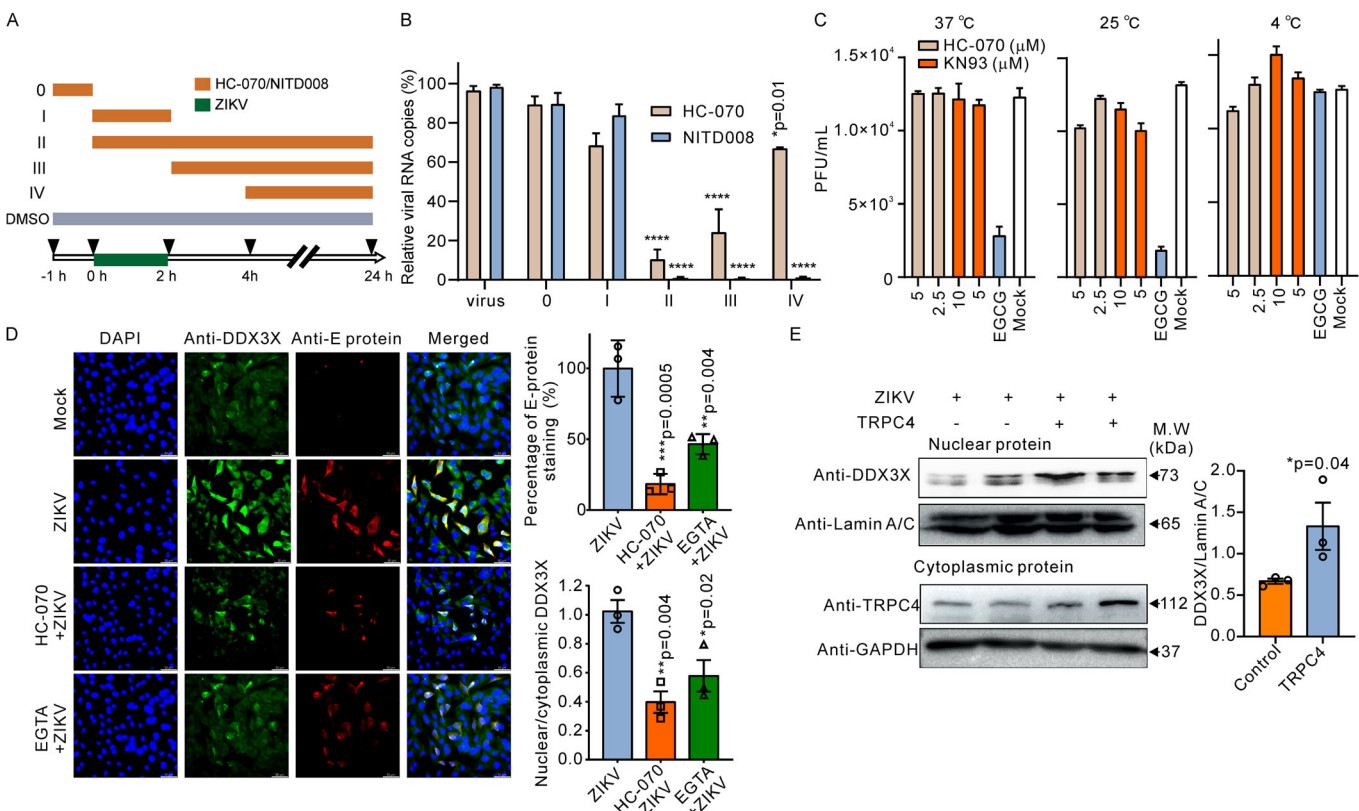

**Figure 5. HC-070 exhibits potential antiviral activity against ZIKV by inhibiting viral replication.**

(A) Timeline of time-of-drug-addition assay. BHK cells were inoculated with the virus during the times indicated with a horizontal green bar (2 h). The orange horizontal bars represent the times of HC-070 (10 μM) or NITD008 (5 μM) treatment. At 24 hpi, total mRNA was extracted for qRT-PCR. BHK cells were infected with ZIKV at a multiplicity of infection of 0.05. The control virus group was treated with DMSO. (B) During the time-of-drug-addition assay, the quantification of viral RNA copies in the total mRNA extracted from cell lysates was performed using qRT-PCR ($n = 4$ biological replicates). (C) The ZIKV ($2 \times 10^6$ PFU) samples were incubated with the indicated compounds or an equal volume of vehicle at 4 °C, 25 °C, and 37 °C for a duration of 1 h. Subsequently, the infectious viral particles in each sample were quantified using the PFU assay ($n = 3$ biological replicates). EGCG was employed as the positive control. Neither HC-070 nor KN-93 exhibited any impact on the infectivity of ZIKV particles in vivo. (D) Representative immunofluorescence images of BHK cells stained with DDX3X (green) and ZIKV E-protein (red) antibodies are shown. The right panel displays graphs with the quantification of data shown in the left panel. The fluorescence intensity of the ZIKV E protein and the nuclear/cytoplasmic DDX3X fluorescence intensity ratio in ZIKV-infected cells exhibited a reduction in both HC-070 and EGTA treatment groups (three independent biological replicates). (E) Representative immunoblot images showing the protein levels of DDX3X and Lamin A/C in the nuclei and TRPC4 and GAPDH in the cytoplasm of ZIKV-infected cells overexpressing TRPC4 versus control ZIKV-infected cells ($n = 3$ biological replicates). The graph in the right panel shows that overexpression of the TRPC4 protein increased the relative level of the DDX3X protein in the nucleus of ZIKV-infected cells. The unpaired T test (one-tailed) and one-way ANOVA followed by the Dunnett's test as the post hoc were employed to determine if there was a significant difference between two groups or among multiple groups, respectively. GAPDH and Lamin A/C staining were used as loading control. Data information: In (B–E), data are presented as mean ± SEM, $*P \leq 0.05$, $**P \leq 0.01$, $***P \leq 0.001$, $****P \leq 0.0001$. Source data are available online for this figure.

groups compared to control until 15 dpi (Fig. 7A). The survival analysis revealed that administration of 0.3 mg/kg HC-070 protected 60% of ZIKV-challenged mice from death, and 3 mg/kg KN-93 protected 30% of ZIKV-challenged mice from death, when none of the ZIKV-challenged mice survived in the vehicle group (Fig. 7B). Importantly, the survival rate of animals in the HC-070-treated group at a dosage of 1.0 mg/kg was observed to be 100%. Furthermore, the administration of HC-070 (1.0 mg/kg) effectively inhibited ZIKV infection in mice, as evidenced by the reduced viral loads observed in the brain of the ICR suckling infection model (Fig. 7C).

We also investigated the impact of HC-070 and KN-93 on the reflexes and behavior of young mice after exposure to ZIKV. However, no differences were found between HC-070, KN-93, and

vehicle-treated ZIKV-infected pups in the grasping reflex test or the hindlimb suspension (HLS) test (Fig. 7D,E).

Spontaneous seizures were reported in ZIKV-infected infants. Therefore, additionally, we determined whether HC-070 or KN-93 can prevent spontaneous seizures in the ZIKV-infected 1-day-old suckling ICR mice. We video-recorded the animals for 1 h per day at 9 dpi, 15 dpi, and 18 dpi. The Racine scale (Table 2) was used to evaluate spontaneous seizures. The average Racine score was significantly reduced in both 0.3 mg/kg and 1 mg/kg HC-070 treated groups at 15 dpi and 18 dpi (Fig. 7F), indicating that the number of seizures was reduced.

Thus, consistent results were obtained in the brain organoid model and the ICR mouse model, namely inhibition of TRPC4 function or decrease in TRPC4 expression by inhibiting CaMKII

**Table 2. Modified Racine scale for visual evaluation of seizures in mice**

| Racine score | Behavioral expression |
|---|---|
| 0 | Normal behavior |
| 1 | Immobilization eye closure, ear twitching, sniffing, facial clonus |
| 2 | Head nodding; mouth and facial movements |
| 3 | Unilateral forelimb clonus |
| 4 | Bilateral forelimb clonus with rearing |
| 5 | Falling, loss of righting reflex accompanied by generalized tonic-clonic seizures |
| 6 | Tonic-clonic seizure extension, possibly leading to death |

successfully alleviated the ZIKV associated seizures and protected neonatal mice from lethal ZIKV challenges.

## Discussion

In this study, we have shown that ZIKV infection increases the host TRPC4 protein expression and that pharmacological inhibition of TRPC4 channels or shRNA-mediated TRPC4 gene knockdown are effective in inhibiting ZIKV replication in host cells. Furthermore, we found that in vivo treatment with a TRPC4 channel blocker or an inhibitor of CaMKII, an upstream protein kinase regulating TRPC4 expression, significantly decreased ZIKV infection-associated seizures and improved the survival of ZIKV-infected neonatal mice. The same inhibitors significantly decreased ZIKV replication in human iPSC-derived brain organoids. These observations together demonstrate that TRPC4 may be one of the key molecules that ZIKV regulates to hijack the host cell calcium signaling system for improving the efficiency of its host cell entry and replication. Our data suggest that TRPC4 may be a key protein underlying some ZIKV infection-induced neuropathological disorders. Remarkably, our findings are consistent with the previous observations that some ion channels may be implicated in neurotropic viral infections (Donate-Macian et al, 2018; He et al, 2020; Hoffmann et al, 2017). The recent outbreak of ZIKV has caused thousands of microcephaly cases and Guillain-Barré syndrome cases, revealing the neurotropic properties of the virus.

We first found a strong correlation between TRPC4 expression and ZIKV's E-protein levels (Fig. 1) in ZIKV-infected cells. Increased level of ZIKV's E proteins were detected in BHK cell overexpressing TRPC4 and elevated amounts of the TRPC4 were detected in cells challenged with increasing ZIKV MOI. We then obtained evidence that the elevated levels of TRPC4 correlate with increased ZIKV infectivity and replication.

We next tested whether the commonly used TRP channel modulators exhibit any anti-Flavivirus activity in BHK cell. Consistent with the previous report by P. Doñate-Macián et al (Donate-Macian et al, 2018), we observed the anti-ZIKV activity of RN-1734, a TRPV4 channel blocker. We next found that HC-070 and ML204, the inhibitors of TRPC4/5, showed significant anti-ZIKV activity in BHK cells. Since AC1903 (Zhou et al, 2017), a specific inhibitor for TRPC5, did not show any anti-ZIKV activity, we focused our investigation on the TRPC4 channel. According to our qRT-PCR results, other TRPC channels (TRPC1/3/5/6) are also

expressed in host cells. However, TRPC1/3/5/6 expression levels were not significantly affected by ZIKV infection.

Unexpectedly, we noticed some anti-ZIKV activity of trivalent cations, $La^{3+}$ and $Gd^{3+}$, which were reported to enhance TRPC4/5 channel activity (Chen et al, 2017) (Table 1). However, $LaCl_3$ showed a broad anti-flavivirus activity. Metal-based nanoparticles have been used as potent antimicrobial agents (Maciuca et al, 2020; Tortella et al, 2021). In this regard, the antiviral activity of $La^{3+}$ or $Gd^{3+}$ may likely be due to its ability to damage the virus directly. Another possibility is that lanthanide ions may make the plasma membrane less susceptible to permeabilization (Gianulis and Pakhomov, 2015; Moulick et al, 2018). $La^{3+}$ may modify the structure of host cell membrane that may prevent viral entry.

We subsequently investigated the ability of ZIKV to regulate TRPC4 channel function and TRPC4 potential contribution to viral replication. The infection of ZIKV is accompanied with fast host intracellular $Ca^{2+}$ increases (Donate-Macian et al, 2018). However, HC-070 did not block ZIKV-elicited initial increases in intracellular $Ca^{2+}$ concentration (Fig. EV2D) during the first hour post infection. This finding indicates that TRPC4 is not responsible for the initial ZIKV-induced intracellular $Ca^{2+}$ rises in infected cells and that those rises may likely drive TRPC4 expression in later stages of the ZIKV life cycle.

It has been demonstrated that ZIKV could modulate CaMKII activity through its NS3 protein, and inhibition of CaMKII significantly suppresses ZIKV infection (Sun et al, 2020). In addition, it was reported that CaMKII plays a key role in regulating TRPC4 expression (Morales et al, 2007). Herein, we did provide evidence that the CaMKII–CREB signaling pathway may be involved in the upregulating of TRPC4 in ZIKV-infected cells during the ZIKV replication stage. The treatment with KN-93, a specific inhibitor of CaMKII, or siRNA-mediated downregulation of CaMKII expression effectively attenuated ZIKV infection (Fig. 4). We also demonstrated herein that the ZIKV-NS3 protein, which reportedly binds and possibly activates CaMKII (Sun et al, 2020), can modulate TRPC4 expression.

Overexpression of ZIKV-NS3 proteins in BHK cells over-expressing the TRPC4 cDNA plasmid (BHK-TRPC4 cells) led to an increased $Ca^{2+}$ influx mediated by TRPC4 channels (Fig. 4). However, ZIKV-NS3 did not increase intracellular $Ca^{2+}$ levels in control BHK cells exhibiting low endogenous TRPC4 expression (Fig. EV2G). We found that overexpression of ZIKV-NS3 augmented TRPC4-mediate $Ca^{2+}$ influx in part by increasing the TRPC4 protein expression rate (Fig. EV2E,F). It was reported that ZIKV-NS3' helicase activity may increase the stability of mRNA (Hooper and Hilliker, 2013). Possibly, ZIKV-NS3 might augment TRPC4 protein expression in BHK-TRPC4 cells by stabilizing TRPC4's mRNA transcribed from the overexpressed TRPC4 cDNA. However, further experiments are needed to prove or refute such a hypothesis.

Our data indicate that ZIKV-NS3 may contribute to the elevation of intracellular $Ca^{2+}$ levels in ZIKV-infected cells in a TRPC4-dependent manner. Since NS3 is produced at the later stages of the ZIKV life cycle, it is not surprising that the augmentation of TRPC4-mediated $Ca^{2+}$ influx occurs only during the replication phase rather than during its initial entry phase. These findings suggest that ZIKV may control TRPC4 function likely via the NS3-CaMKII pathway. However, we cannot rule out the existence of additional mechanisms that may facilitate the ZIKV ability to hijack the host's $Ca^{2+}$ signaling

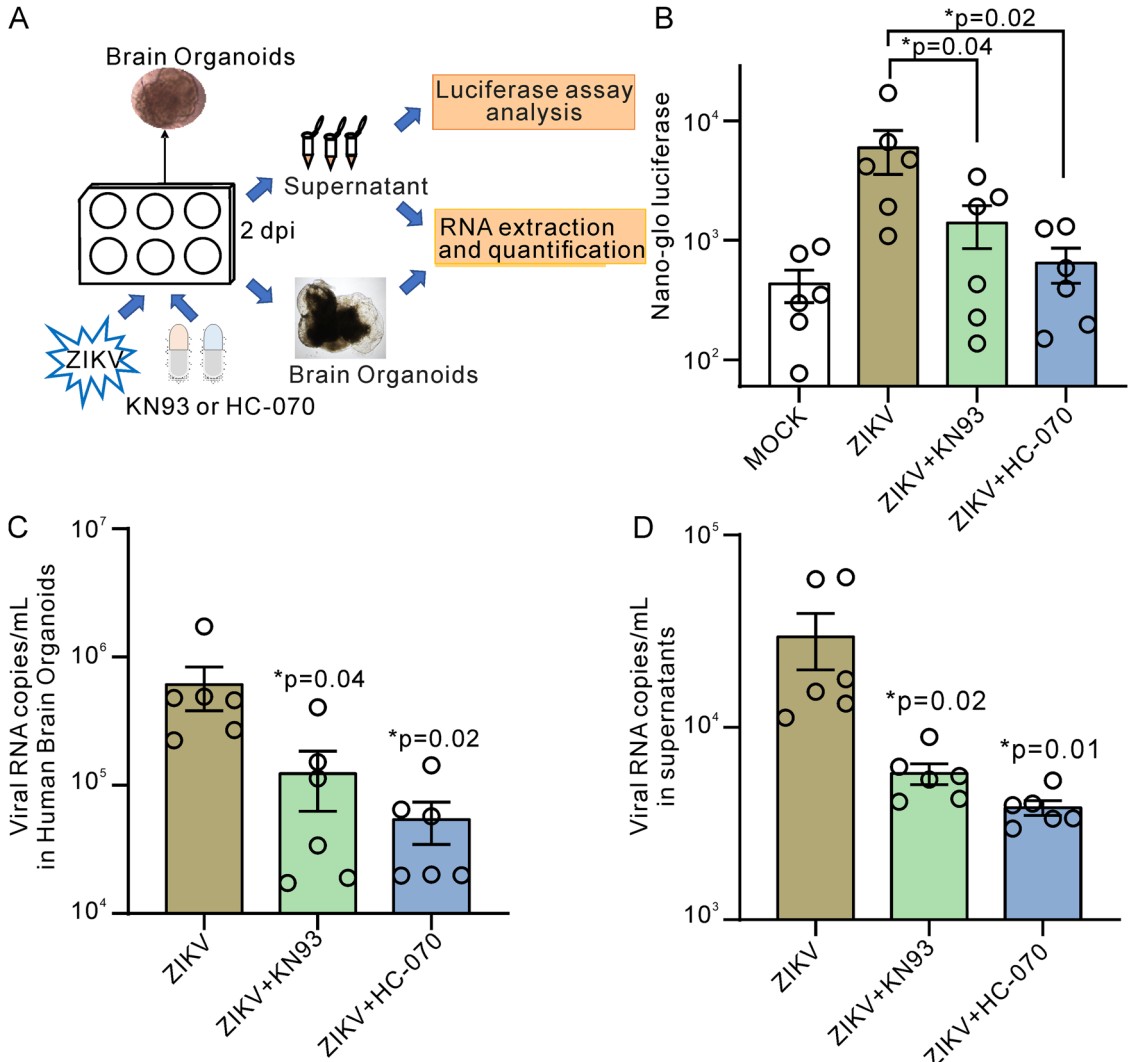

**Figure 6. The protective efficacy of HC-070 and KN-93 in human brain organoids infected with ZIKV.**

(A) Design of experiments involving human brain organoids. Human brain organoids were infected with $1.2 \times 10^6$ PFU ZIKV or mock medium at 30 days. At 2 dpi, the supernatant was collected for the luciferase assay and viral content assay, and organoids were used to extract viral RNA. (B) Luciferase Assay. The luminescence was measured as described in the Method section in the presence or absence of 10 μM HC-070 or 10 μM KN-93 ($n = 6$ brain organoids). (C, D) Viral RNA analyses by qRT-PCR. The viral RNA copy number was determined in the supernatant and organoids of each tested group ($n = 6$ brain organoids) and it was decreased in the presence of HC-070 and KN-93. The one-way ANOVA test followed by the Dunnett's post hoc test was employed to determine if there was a significant difference among multiple groups. Data information: In (B–D), data are presented as mean ± SEM, *$P \leq 0.05$. Source data are available online for this figure.

machinery. There are yet unresolved complexities surrounding the intricate mechanisms by which ZIKV manipulates the host's calcium signaling system.

Increased intracellular $Ca^{2+}$ levels are important for ZIKV replication (Bezemer et al, 2022). Upregulated $Ca^{2+}$-permeable TRPC4 channels can indeed contribute to increased intracellular $Ca^{2+}$ levels, thereby potentially promoting the replication of ZIKV possibly by facilitating DDX3X translocation to the nuclei of ZIKV-infected cells. In fact, inhibiting TRPC4 activity with HC-070 or reducing extracellular $Ca^{2+}$ levels with EGTA did decrease DDX3X nuclear localization in ZIKV-infected BHK cells (Fig. 5D).

Remarkably, Vinayagam et al, reported that $Ca^{2+}$/calmodulin binding to the rib helix of TRPC4 stabilizes the channel in its closed conformation (Vinayagam et al, 2020), suggesting that $Ca^{2+}$/calmodulin binding may negatively regulate TRPC4 activity. Conversely, Shaefer et al, found that intracellular $Ca^{2+}$ increases are critical for TRPC4 function and that TRPC4 cannot be activated if intracellular $Ca^{2+}$ is chelated with 10 mM EGTA (Schaefer et al, 2000). Consistently, Duan et al, revealed that the TRPC4 protein contains a separate $Ca^{2+}$ binding site located in a pocket on the cytoplasmic side of the S1–S4 domain (Duan et al, 2018). Thus, ZIKV-induced intracellular $Ca^{2+}$ rises at the replication stage may also likely potentiate TRPC4 activity, besides upregulating TRPC4 expression.

Seizures have been reported after ZIKV infection in adult patients, and studies have shown that 50% to 60% of microcephalic

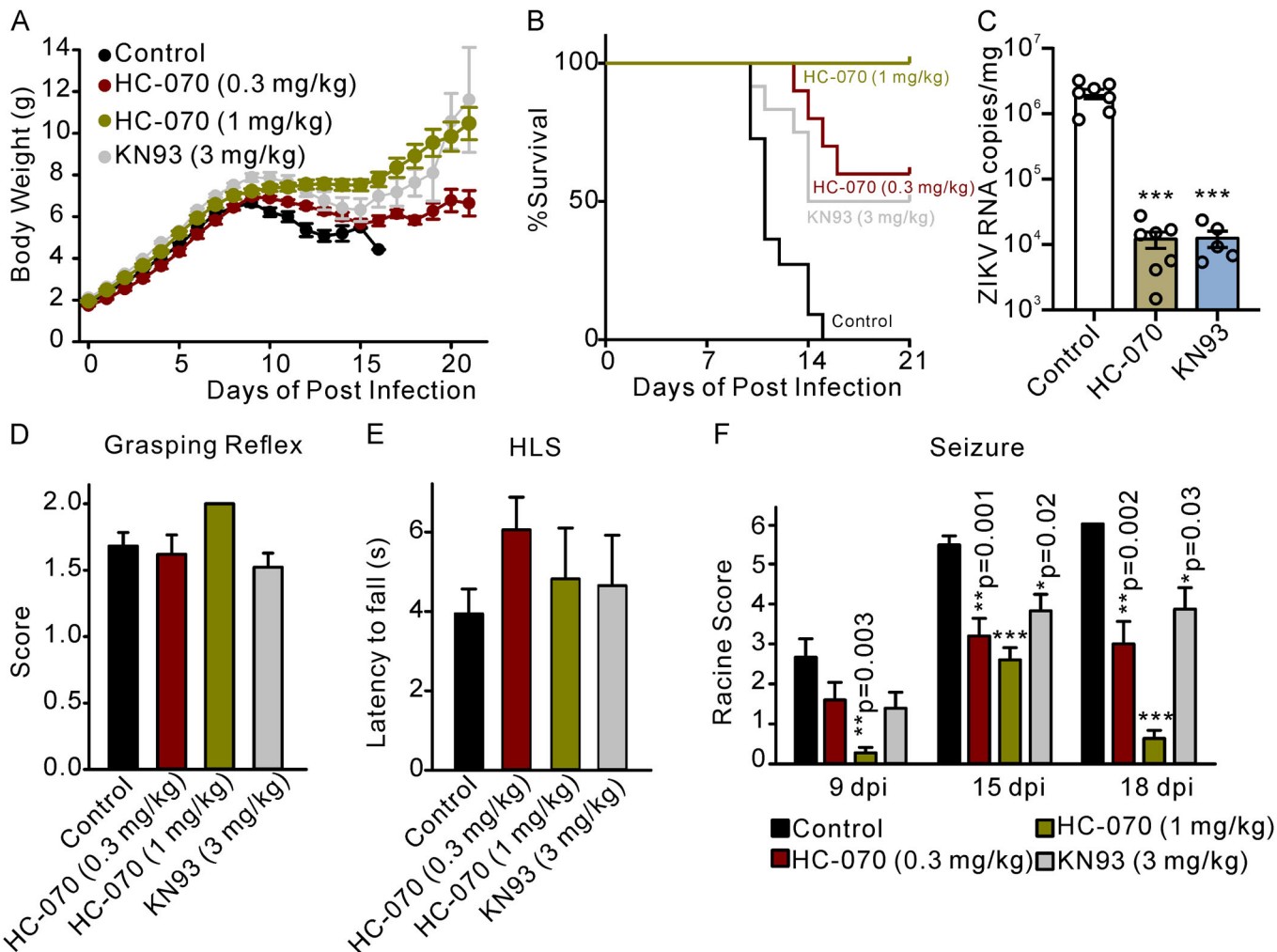

**Figure 7. HC-070 and KN-93 reduce seizures and improve the survival in ZIKV-infected ICR mice.**

Mice were injected intraperitoneally with $2 \times 10^4$ PFU ZIKV at post-natal day 1, and HC-070 (0.3 mg/kg and 1 mg/kg), KN-93 (3 mg/kg), or vehicle was IP injected starting at 2 dpi and then for 10 consecutive days. (A, B) The graphs show body weight changes and animal survival curves ($n = 10$–20 mice). HC-070 and KN-93 treatments markedly improved the survival of ZIKV-infected mice. (C) The viral loads in the brain were quantified using qRT-PCR ($n = 5$–7 biological replicates). HC-070 and KN-93 treatments decreased ZIKV RNA copies. (D–F) The grasping reflex (D) was assessed in each group of mice at 7 dpi ($n = 10$–20 mice). Subsequently, the latency to fall in the hindlimb suspension test was measured at 9 dpi (E), $n = 10$–20 mice. In addition, Racine scores were recorded to evaluate seizures in ZIKV-infected animals (F). The one-way ANOVA test followed by the Dunnett's post hoc test was employed to determine if there was a significant difference among multiple groups. Data information: In (A, C–F), data are presented as mean ± SEM, $*P \leq 0.05$, $**P \leq 0.01$, $***P \leq 0.001$. Source data are available online for this figure.

babies owing to ZIKV congenital infection develop spontaneous epileptic activity. Recent studies with genetic ablation of various TRPC family members have demonstrated that TRPC channels play a critical role in pilocarpine-induced acute seizures (Phelan et al, 2012; Zheng and Phelan, 2014). The epileptiform activity induced by mGluR agonists is significantly reduced in TRPC4-KO mice. Therefore, the increased expression of TRPC4 may be a key step underlying seizure induced by ZIKV infection. Since ZIKV is quickly cleared in wild-type adult mice infected by ZIKV, we used the 1-day-old sucking ICR mouse model (Nem de Oliveira Souza et al, 2018; Yang et al, 2020) to test the hypothesis that HC-070 and KN-93 may protect against ZIKV infection. It has been reported that during the acute phase of infection with ZIKV, the infected newborn wild-type mice develop frequent seizures (Nem de Oliveira Souza et al, 2018). No differences were found among

vehicle, HC-070, and KN-93 pretreated ZIKV-infected pups in the grasping reflex test and the hindlimb suspension (HLS) test. Seizures were evaluated using the Racine score at 9 dpi, 15 dpi, and 18 dpi and reached higher scores at 15 dpi and beyond (Fig. 7F). The administration of HC-070 or KN-93 resulted in a significant reduction in seizure scores and significantly improved survival rates in neonatal ZIKV-infected mice, demonstrating the efficacy of these in vivo treatments. Notably, no mortality was observed within the 1.0 mg/kg HC-070 treated group. Thus, the downregulation of TRPC4 channel function may be an effective strategy for treating seizure disorders associated with ZIKV infection.

Investigations of neurotropic virus infections are usually performed in cell lines and animal models. Due to the disadvantages of animal models, human-derived brain models have been established to study mechanisms of neurotropic virus-induced

neural disorders. Human iPSC-derived brain organoids have already been used to investigate the infection mechanism of ZIKV. Here we used the human iPSC-derived brain organoids model to test the therapeutic efficacy of HC-070 and KN-93. We found that HC-070 or KN-93 significantly inhibited the replication of ZIKV in human iPSC-derived brain organoids (Fig. 6), which agrees with the results obtained in our ICR mouse model.

This study has some limitations and weaknesses. The infection of various viruses, including ZIKV, is accompanied by alterations in $Ca^{2+}$ signaling within the host cell, which constitutes an intricate process involving multiple components of the host $Ca^{2+}$ signaling system. The ability of ZIKV to modulate TRPC4 expression and function via the ZIKV-NS3–CaMKII–CREB pathway, which we describe herein, constitutes a very plausible mechanism. However, there may be many other cellular pathways and elements promoting ZIKV infection, and our current study does not aim at uncovering every possible pathway or element involved in the ZIKV pathogenesis. Thus, further experiments will be needed to better understand the role of $Ca^{2+}$ signaling during ZIKV infection.

TRPCs are implicated in the continually growing number of cell functions (Chen et al, 2020). Activated TRPCs allow the influx of $Ca^{2+}$ and monovalent alkali cations into the cytosol of cells, leading to cell depolarization and rising intracellular $Ca^{2+}$ concentration. Here, we reveal a critical role of TRPC4 during ZIKV infection. We provide evidence that ZIKV infection may trigger intracellular $Ca^{2+}$ increases to promote TRPC4 expression via the activation of CaMKII–CREB, TRPC4-mediated $Ca^{2+}$ influx in turn increases nuclear translocation of DDX3X, a critical step during ZIKV replication, and ZIKV-NS3 potentiates TRPC4 protein expression and function, further stimulating ZIKV replication. All these signaling events may lead to seizures and death in neonatal ICR mice. Thus, TRPC4 channel inhibitors or CaMKII antagonists may be useful in reducing ZIKV infection and resulting neurological dysfunctions. Our findings presented herein will help in developing novel approaches for preventing ZIKV infection and reveal that the TRPC4 channel is a novel potential target for anti-ZIKV drug discovery efforts.

# Methods

### Reagents and tools table

| Reagent/resource | Reference or source | Identifier or catalog number |
| --- | --- | --- |
| **Experimental models** | | |
| BHK cells (*Mesocricetus auratus*) | National Infrastructure of Cell Line Resource | 1101HAM-PUMC000097 |
| U87 cells (*Homo sapiens*) | National Infrastructure of Cell Line Resource | 1101HUM-PUMC000208 |
| Huh7 cells (*Homo sapiens*) | National Infrastructure of Cell Line Resource | 1101HUM-PUMC000679 |
| A172 cells (*Homo sapiens*) | National Infrastructure of Cell Line Resource | 1101HUM-PUMC000941 |
| H4 cells (*Homo sapiens*) | National Infrastructure of Cell Line Resource | 1101HUM-PUMC000372 |
| Vero cells (*Chlorocebus sabaeus*) | American Type Culture Collection | CCL-81 |

| Reagent/resource | Reference or source | Identifier or catalog number |
| --- | --- | --- |
| HT22 cells (*Mus musculus*) | MERCK | N/A |
| Specific-pathogen-free (SPF) ICR mice | Beijing Vital River Laboratory Animal Technology Co. | N/A |
| A129 mice (*Ifnar$^{-/-}$*) | State Key Laboratory of Pathogen and Biosecurity | N/A |
| **Recombinant DNA** | | |
| pEnCMV-TRPC1-3×FLAG | Miaoling plasmid sharing platform | NM_001251845.2 |
| pEnCMV-TRPC4-m-FLAG | Miaoling plasmid sharing platform | NM_016179.4 |
| pEnCMV-TRPC5-m-FLAG | Miaoling plasmid sharing platform | NM_009428.3 |
| pEnCMV-TRPC6-3×FLAG | Miaoling plasmid sharing platform | NM_013838.2 |
| pcDNA3.1(+)-ZIKV-NS3(co_Human)*(ns)/FLAG | VectorBuilder | N/A |
| pcDNA3.1(+)-ZIKV-NS3(co_Human)*(ns)/Myc/6*his | VectorBuilder | N/A |
| **Antibodies** | | |
| Anti-TRPC4 antibodies | Cell Signaling Technology | Cat # ab83689 |
| Monoclonal mouse anti-GAPDH antibody | Abcam | Cat # ab8245 |
| Monoclonal mouse anti-α-tubulin antibody | Abcam | Cat # ab7291 |
| Horseradish peroxidase (HRP)-conjugated goat anti-mouse IgG antibody | Abcam | Cat # ab19195 |
| Monoclonal anti-flavivirus group antigen–antibody | Merck-Millipore | Cat # MAB10216 |
| Anti-GFAP antibody | Servicebio | Cat # GB12096 |
| Anti-NeuN antibody | Cell Signaling Technology | Cat # GB12096 |
| Anti-CREP antibody | BioFront Technologies | Cat # 9197 |
| Anti-pCREP antibody | BioFront Technologies | Cat # 9198 |
| Anti-ZIKV-NS1 monoclonal antibody | GeneTex | Cat # BF-1225-06 |
| Anti-ZIKV-NS5 monoclonal antibody | GeneTex | Cat # GTX133329 |
| Anti-ZIKV E-protein antibody | Merck-Millipore | Cat. No. MAB10216-I |
| Zika virus Envelope protein antibody | GeneTex | Cat # GTX133314 |
| DDX3 Polyclonal antibody | Proteintech | Cat # 11115-1-AP |
| Lamin A/C Polyclonal antibody | Proteintech | Cat # 10298-1-AP |
| CaMKII Gamma Polyclonal antibody | Proteintech | Cat # 12666-2-AP |
| Alexa Fluor 488 donkey anti-mouse IgG (H + L) | Thermo Fisher Scientific | Cat # A-21202 |
| **Oligonucleotides and other sequence-based reagents** | | |
| siRNA-CaMKII | This study | Methods |
| siRNA-CREB | This study | Methods |

| Reagent/resource | Reference or source | Identifier or catalog number |
|---|---|---|
| ZIKV forward | This study | Methods |
| ZIKV reverse | This study | Methods |
| TRPC4 forward | This study | Methods |
| TRPC4 reverse | This study | Methods |
| TRPC1 forward | This study | Methods |
| TRPC1 reverse | This study | Methods |
| TRPC3 forward | This study | Methods |
| TRPC3 reverse | This study | Methods |
| TRPC5 forward | This study | Methods |
| TRPC5 reverse | This study | Methods |
| TRPC6 forward | This study | Methods |
| TRPC6 reverse | This study | Methods |
| ZIKV-NS5 forward | This study | Methods |
| ZIKV-NS5 reverse | This study | Methods |
| ZIKV-NS5 probe | This study | Methods |
| **Chemicals, enzymes, and other reagents** | | |
| HC-070 | MedChemExpress | Cat # HY-112302 |
| KN-93 | Selleck | Cat # S6787 |
| NITD008 | MedChemExpress | Cat # HY-12957 |
| EGCG | MedChemExpress | Cat # HY-13653 |
| EGTA | MedChemExpress | Cat # HY-D0861 |
| Triton X-100 | Molecular Dimensions | Cat # MD12-603 |
| Hoechst 33342 Fluorescent Stain Solution | Thermo Fisher | Cat # H21492 |
| **Software** | | |
| SigmaPlot 12.5 software | Grafiti LLC | Version 12.5 |
| **Other** | | |
| Minimum Essential Medium | Gibco | Cat # 11095080 |
| Dulbecco's Modified Eagle Medium | Gibco | Cat # 11965118 |
| fetal bovine serum | Gibco | Cat # A5670701 |
| penicillin (100 U/mL)-streptomycin | Invitrogen | Cat #15140-122 |
| STEMdiff™ Cerebral Organoid Kit | STEMCELL Technologies | Cat # 08570 |
| Nano-Glo Luciferase Assay | Promega | Cat # N110 |
| CellTiter-Glo Luminescent Cell Viability Assay | Promega | Cat # G7573 |
| Lipofectamine 3000 reagent | Invitrogen | Cat # L3000150 |
| RNeasy mini kit | QIAGEN | Cat # 74104 |
| One Step PrimeScript RT-PCR Kit | TaKaRa | Cat #RR64JW |
| Pierce® IP lysis buffer | Thermo Scientific | Cat # 87787 |
| FLAG antibody and protein A/G agarose beads | Beyotime | Cat # P2202M |

## Cell lines and virus

The BHK, U87, Huh7, A172, and H4 cell line were obtained from the National Infrastructure of Cell Line Resource, China. Vero cells and HT22 cells were purchased from the American Type Culture Collection (ATCC) and MERCK, respectively. The U87 cells were cultured in Minimum Essential Medium (MEM), while the other cell lines were cultivated in Dulbecco's Modified Eagle Medium (DMEM; Gibco), supplemented with 10% fetal bovine serum (FBS, Gibco) and penicillin (100 U/mL)-streptomycin (100 μg/mL; Invitrogen) at 37 °C in 5% $CO_2$. ZIKV (SMGC-1 strain) was produced by infecting Vero cells 3 days post-seeding at a multiplicity of infection (MOI) of 0.001. The virus titers were determined by the PFU assay or 50% tissue culture infective dose infectivity assay ($TCID_{50}$). The same volume of virus-free conditioned medium of VERO cells was used as a mock control.

## Animals and neonatal infection

Specific-pathogen-free (SPF) ICR mice used in this research were purchased from Beijing Vital River Laboratory Animal Technology Co., Ltd. The A129 mice ($Ifnar^{-/-}$; gifts from State Key Laboratory of Pathogen and Biosecurity, Beijing Institute of Microbiology and Epidemiology) were bred in the animal laboratory of the Beijing Institute of Pharmacology and Toxicology. All experiments performed in mice followed the protocols and procedures reviewed and approved by the Animal Experiment Committee of the Laboratory Animal Center of the Beijing Institute of Pharmacology and Toxicology (IACUC-DWZX-2022-604). All experiments with infectious virus were performed in a biosafety level 2 (BSL-2) or animal biosafety level 2 laboratory (ABSL-2). ICR suckling mice were infected intraperitoneally with $2 \times 10^4$ PFU ZIKV or the same volume of mock medium. All littermates received the same treatment to avoid cross-contamination. Mice showing any signs of reflux or hemorrhage were excluded from further analyses. HC-070 (0.3 mg/kg/day) or KN-93 (3 mg/kg/day) was intraperitoneally administered at 2 dpi and then daily for 10 consecutive days. Pups were evaluated daily for mortality, and body weight was measured every day up to 21 dpi. A separate subset of mice was euthanized at 12 dpi, and the brain hemispheres were collected for RNA isolation and immunofluorescence staining. $Ifnar^{-/-}$ A129 mice were infected with ZIKV ($2 \times 10^4$ PFU) via retro-orbital injection.

## Brain organoid cultures

Generation and cultivation of cortical organoids from human iPSCs were performed according to the protocol of STEMdiff™ Cerebral Organoid Kit (STEMCELL Technologies) (Jacob et al, 2020). On day 30, brain organoids were infected in a 12-well plate (Corning) with $1.2 \times 10^6$ PFU of ZIKV in the presence of vehicle, HC-070, or KN-93. At 2 dpi, the supernatant was collected to measure the luminescence using the Nano-Glo Luciferase Assay (Promega; cat #N110). mRNA was extracted from the collected organoids, and ZIKV RNA levels were quantified by performing qRT-PCRs.

## Chemicals and reagents

All test compounds were purchased from Selleck or MedChemExpress and dissolved in DMSO at 100 mM stock concentration for in vitro tests. The anti-TRPC4 antibodies were obtained from Cell Signaling Technology (cat # ab83689) and Alomone labs (cat # CC-018), The monoclonal mouse anti-GAPDH antibody (cat #ab8245), monoclonal mouse anti-α-tubulin antibody (cat #ab7291), and horseradish peroxidase (HRP)-conjugated goat anti-mouse IgG antibody were purchased from Abcam. The monoclonal anti-flavivirus group antigen–antibody (clone D1-4G2-4-15, protein E; cat #MAB10216, anti-E protein), anti-GFAP (cat# GB12096), anti-NeuN (cat # GB12096), anti-CREP (cat #9197), anti-pCREP (cat #9198), anti-ZIKV-NS1 monoclonal antibody (cat # BF-1225-06), and anti-ZIKV-NS5 monoclonal antibody (GeneTex, cat # GTX133329) were purchased from Merck-Millipore, Servicebio, Cell Signaling Technology, GeneTex, and BioFront Technologies, respectively. The anti-ZIKV E-protein antibody from Merck-Millipore (Cat. No. MAB10216-I) was used in the immunofluorescence experiments. According to the manufacturer's recommendations, this antibody is not suitable for WB. Therefore, the anti-ZIKV-NS1 antibody was employed for western blots, and Zika virus Envelope protein antibody (GTX133314) was used for Co-IP experiment. The antibodies against DDX3X, Lamin A/C, and CaMKII were 11115-1-AP, 10298-1-AP, and 12666-2-AP, respectively. Alexa Fluor 488 donkey anti-mouse IgG (H + L) (cat # A-21202) and Hoechst 33342 Fluorescent Stain Solution (cat # H21492) were obtained from Thermo Fisher Scientific.

## Cell survival

Confluent BHK cells cultured in 96-well plates were infected with $100 \times TCID_{50}$ ZIKV in the presence of either HC-070 or KN-93 for the indicated number of days of infection. The protective effect of HC-070 and KN-93 was assessed using the CellTiter-Glo Luminescent Cell Viability Assay (Promega, USA) according to the manufacturer's instructions. Values were calculated using SigmaPlot 12.5 Software.

## Cell transfection

BHK cells were transfected using the Lipofectamine 3000 reagent (Invitrogen, Carlsbad, CA, USA) in accordance with the manufacturer's recommendations. The TRPCs (with GenBank accession numbers NM_001251845.2, NM_016179.4, NM_009428.3, and NM_013838.2 for TRPC1, TRPC4, TRPC5, and TRPC6, respectively) plasmids were obtained from Miaoling plasmid sharing platform (Wuhan, Hubei, China). ZIKV-NS3 (ZIKV, SMGC-1 strain) plasmid were custom-synthesized by VectorBulider (Guangzhou, China) in the pcDNA3.1+ vector. The R-GECO Red Fluorescent $Ca^{2+}$ Assay was purchased from Montana Molecular (#U0600R, Bozeman, MT, USA), and the BHK cells were transfected with the R-GECO sensor plasmid according to the manufacturer's instructions. The cells were cultured for 24–48 h before the western blots or the fluorescence imaging experiments were performed.

## RNAi

shRNA lentiviral particles from Santa Cruz (sc-42669-V) were used for the downregulation of TRPC4 in BHK cells following the

manufacturer's instructions. The oligonucleotides targeting CaMKII and CREB (5'-GGGAUGAGGAUCAGCACAATT-3' and 5'-CUGGA-GACGUACAAACAUATT-3', respectively) were obtained from Shanghai Genechem Co., Ltd. shRNA or siRNA treatment was carried out for 48 h. Cells were infected with ZIKV and then incubated for another 48 h. mRNA was then extracted using the RNeasy mini kit (QIAGEN, USA) to measure viral RNA levels by qRT-PCR as described below.

## Quantitative real-time PCR (qRT-PCR)

The total mRNA from infected BHK cells, animal brains, or human brain organoids was harvested using the RNeasy mini kit (QIAGEN, USA). Five hundred nanograms of RNA was used in reverse transcription reactions (One Step PrimeScript RT-PCR Kit, TaKaRa, Japan). Real-time PCR was performed using SYBR Green (Roche). Levels of mRNA expression were normalized to the expression of GAPDH as an internal control and were expressed relative to its mean values seen in samples. Primers used for PCR were as follows: ZIKV (the linker region of membrane protein and envelope protein) forward: TTGGTCATGATACTGCTGATTGC, ZIKV reverse: CCYTCCACRA AGTCYCTATTGC; TRPC4 forward: GATGAACTCCTTGTATCTG, TRPC4 reverse: GGCTTTGACATTGGTAAC; TRPC1 forward: CAA-GAT TTTGGGAAATTTCTGG, TRPC1 reverse: TTTATCCTCAT-GATTTGCTAT; TRPC3 forward:TGACTTCCGTTGTGCTCAAATA TG, TRPC3 reverse: CCTTCTGAAGCCTTCTCCTTCTGC; TRPC5 forward: ATCTACTGCCTAGTACTACTGGCT, TRPC5 reverse: CAGCATGATCGGCAATGAGCTG; TRPC6 forward: AAAGATAT CTTCAAATTCATGGTC, TRPC6 reverse: CACGTCCGCATCATCC TCAATTTC. In the case of human brain organoids, primers and probe specific for ZIKV-NS5 were as follows. Forward primer was GGTCAG CGTCCTCTCTAATAAACG; reverse primer was GCACCCTAGTGT CCACTTTTTCC; probe was AGCCATGACCGACACCACACCGT. Virus RNA copy numbers were calculated using the ΔCT method, with the cycle threshold (Ct) value being obtained for each sample and compared to the known copy number standard curve.

## Intracellular calcium measurement

Cells transfected with the R-GECO plasmid were examined using an Inverted Fluorescence Microscope (IX73-OLYMPUS) approximately 36-48 h post transfection. The intracellular $Ca^{2+}$ levels were then quantified. For the experimental groups subjected to ZIKV virus infection, the assessment of red fluorescence intensity, indicative of an intracellular $Ca^{2+}$ level increase, was conducted employing a multimode reader (Thermo Scientific™ Varioskan™ LUX, Ex 550 ± 20 nm, $E_m$ 610 ± 20 nm for R-GECO). For the other groups of cells, intracellular $Ca^{2+}$ dynamics were monitored utilizing the Cell Imaging Multimode Reader (BioTek Cytation 5, Alilent) at a controlled temperature of 37 °C and in a 5% $CO_2$ environment. Live-cell imaging was carried out using a 10× objective lens with a numerical aperture of 0.5. The Gene 5 analysis system was subsequently employed to compute the ratio of red fluorescence signals to the baseline fluorescence signal, serving as an accurate indicator of $Ca^{2+}$ signaling dynamics.

## Nuclear protein extraction

Cells from different treated groups were resuspended in a lysis buffer (containing 10 mM HEPES pH 7.4, 10 mM KCl, 0.05% NP-

40, and protease inhibitors) and incubated on ice for 20 min. The samples were then subjected to centrifugation at 14,000 rpm for 10 min at a temperature of 4 °C. Supernatants were collected and used as the cytoplasmic fraction. The remaining pellets were washed with the lysis buffer and underwent another round of centrifugation under the same conditions mentioned above. As a result, the supernatants were removed while the pellets obtained represented the nucleoplasmic fraction. These nucleoplasmic fractions were subsequently resuspended using the lysis buffer and completely disrupted through sonication. Both cytoplasmic and nucleoplasmic proteins were quantified by BCA assay before being analyzed via western blots.

## Co-IP assay

The Co-immunoprecipitation (Co-IP) of ZIKV-E protein and TRPCs was performed using the lysates of BHK cells. Cells were transfected with either an empty vector (FLAG) or plasmids of FLAG-tagged TRPC4, TRPC1, TRPC5, and TRPC6 using Lipofectamine® 3000 reagent. After 24 h post transfection (p.t.), the cells were infected with ZIKV at a MOI of 0.001. At 48 h post transfection or 24 h post-ZIKV infection, the cells were resuspended in ice-cold Pierce® IP lysis buffer (Thermo Scientific) containing phosphatase inhibitors (NaF and Na3VO4). Immunoprecipitation was carried out overnight at 4 °C using FLAG antibody and protein A/G agarose beads (Beyotime). The beads were washed six times, and the precipitated proteins were analyzed by immunoblotting using anti-E protein and anti-flag antibodies.

For the Co-IP of ZIKV-NS3 protein and CaMKII, BHK cells were transfected with an empty vector (Myc) or plasmids encoding Myc-NS3 using Lipofectamine® 3000 reagent. At 48 h post transfection, the cells were resuspended in ice-cold Pierce® IP lysis buffer (Thermo Scientific) containing phosphatase inhibitors (NaF and Na$_3$VO$_4$). Immunoprecipitation was performed overnight at 4 °C using either anti-CaMKII antibody or anti-NS3 antibody along with protein A/G agarose beads from Beyotime. Beads underwent six washes before analyzing the precipitated proteins through immunoblotting with either anti-NS3 or anti-CaMKII antibodies.

## Immunoblotting

Confluent cells seeded in 12-well plates were inoculated with ZIKV (SMGC-1 strain) at a MOI of 0.001. At 72 hpi, cells were washed twice with PBS and lysed in 300 μL of RIPA lysis buffer with protease and phosphatase inhibitor cocktail (Invitrogen). Equal amounts of protein extracts (50 μg) were resolved by 4 to 15% SDS/PAGE and transferred to polyvinylidene difluoride membranes (Millipore). Membranes were blocked in blocking buffer (5% non-fat-dry milk TTBS 1×) at 4 °C. Primary antibodies were incubated in blocking buffer for 1 h at room temperature or overnight at 4 °C. Primary antibodies were diluted in blocking buffer at the following ratios: anti-NS1 (1:1000), anti-TRPC4 (1:200) and anti-GAPDH or anti-tubulin (1:1000). Secondary antibodies were incubated for 1 h at room temperature. Anti-mouse (GE Healthcare) and anti-rabbit (GE Healthcare) secondary antibodies were used at a 1:5000 dilution in blocking buffer. Protein lysates were analyzed by western blot and imaged with a BIO-RAD Gel Doc XR + Imaging system (Bio-Rad).

## Immunofluorescence assay

ZIKV-infected cells were fixed with 4% paraformaldehyde at room temperature and washed with PBS. Isolated mouse brains were fixed in 4% paraformaldehyde for 4 h on ice. The tissue was then washed with PBS, and frozen in Tissue-Tek O.C.T. Compound (Sakura Finete) using a mixture of dry ice and isopentane. After cryosections (~ 8 μm) were obtained, O.C.T. was removed from tissue samples by a 5–10 min wash in Tris-buffered saline (TBS). Samples were permeabilized in 0.2% Triton X-100 in TBS for 5 min and were then washed three times in TBS. After that, cells or sections were blocked with 5% BSA and incubated with the primary antibody (mouse anti-flavivirus group antigen–antibody, anti-TRPC4 antibody, or anti-DDX3X antibody) at 37 °C for 1 h. Then, cells were washed again and incubated with a secondary antibody for 1 h at 37 °C. The cells were then washed, and the nuclei were stained with the Hoechst 33342 fluorescent stain for 30 min at room temperature. The digital tissue section scanner (Pannoramic 250 FLASH, 3DHISTECH) was utilized for imaging brain tissue slides, while the High-Content Imaging System (DMi8 automated, LEICA) or Laser confocal microscopy (OLYMPUS FV3000, OLYMPUS) were employed to perform fixed cell analysis in plates with objectives of 40× (dry lens) or 60× (oil lens) with the numerical apertures of 0.5 and 1.4, respectively.

## Time-of-drug-addition assay

Time-of-drug-addition assay was performed as previously described with some modifications (Yan et al, 2022). Briefly, confluent BHK cells plated at a 12-well plate ($3 \times 10^5$ cells/well) were cultured at 37 °C overnight. Cells were then incubated with ZIKV inoculum (MOI of 0.05) between 0 and 2 h, and HC-070 (10 μM) or NITD008 (positive control drug, 5 μM) were added at four different intervals that represent various stages of the viral infection cycle. The inoculum was removed at the end of each stage, and the cells were washed with medium three times. After 24-h of treatment, the total mRNA was extracted and then used for qRT-PCR assays.

## Infectivity inhibition assay

The virus solution was treated with equal amounts of HC-070 (2.5 and 5 μM), KN-93 (5 and 10 μM), or EGCG (25 μM, used as a positive control) at various temperatures for 1 h to assess the impact of HC-070 or KN-93 on ZIKV infectivity. The resulting mixture was then quantified using the PFU assay after being diluted and incubated with Vero cells for 2 h. Following removal of the inoculum, the cells were washed thrice with PBS and subsequently covered with DMEM containing 1.5% low-melting point agarose (Promega). After 96 h, the cells were fixed using a solution of 4% paraformaldehyde and stained utilizing a dye solution consisting of 1% crystal violet. Finally, plaque counting was performed.

## Seizure evaluation

Seizure evaluation was performed following the protocol described elsewhere (Nem de Oliveira Souza et al, 2018). Recordings of seizures were performed simultaneously in all groups for 1 h daily. Seizure recordings were always performed between 11 am and

### The paper explained

#### Problem

Zika virus (ZIKV) infection poses a significant threat to global health, particularly due to its association with severe neurological outcomes such as seizures and early infancy death. Since there is currently no specific antiviral therapy available against ZIKV, it is imperative to identify new therapeutic targets to supplement existing supportive treatment strategies. Thus, understanding the mechanisms underlying ZIKV pathogenesis, especially in the context of neurological damage, remains a critical challenge.

#### Results

This study explores the role of transient receptor potential canonical (TRPC) channels, known for their involvement in nervous system excitability and seizure development, in the pathogenesis of ZIKV infection. Our research demonstrates that ZIKV infection leads to an increase in TRPC4 expression within host cells. The upregulation occurs when the ZIKV-NS3 protein interacts with Calcium/Calmodulin-dependent protein kinase II (CaMKII), enhancing TRPC4-mediated calcium influx which is crucial for viral replication and spread. Importantly, the application of inhibitors targeting TRPC4 or CaMKII was found to reduce seizure occurrences and increase survival rates in neonatal mice infected with ZIKV. These inhibitors also effectively blocked the spread of ZIKV in brain organoids derived from human-induced pluripotent stem cells, highlighting their potential as a novel therapeutic approach.

#### Impact

The findings of this study elucidate the pivotal role played by the CaMKII–CREB pathway and TRPC4 expression in ZIKV pathogenesis, offering novel avenues for anti-ZIKV therapies. By highlighting TRPC4 as a potential pharmacological target, this research not only advances our understanding of ZIKV's neurological impact but also unveils promising strategies to mitigate its deleterious effects on the nervous system.

midday in the tested animals. For seizure recordings, dams and their respective pups were removed from their home cages and placed in a box measuring 41 cm × 34 cm × 18 cm containing clean sawdust and high-resolution recording was performed using a Nikon D3300 camera fixed on the ceiling of the room. Seizure Video recordings were analyzed by an experienced researcher blinded to the experimental condition, who identified animals showing any degree of modified Racine scale seizures (Table 2).

### Grasping reflex

Temporal development of the grasping reflex was evaluated in newborn mice from the day of infection to 7 dpi. Mice were held by the scruff of their necks and each front paw was individually stroked with the blunt end of a small paperclip. Scores were given as such: 2 if the reflex is present in both front paws; 1 if the reflex is present in only one front paw; 0 if the reflex is absent in both front paws.

### Hindlimb suspension

At 9 dpi, animals were individually removed from their home cages and placed inside a 50 mL laboratory tube padded with cotton balls, facing the interior of the tube, and suspended by the hindlimbs.

Time to fall inside the tube was measured as an assessment of muscle strength and general neuromuscular function.

### Statistical analysis

Each experiment was performed with at least three replicates. Data shown are the average of at least three independent experiments, or as indicated in the figure legends. Statistical analysis was performed using SigmaPlot 12.5 software. Unpaired $T$ tests (two-sided) or one-way ANOVA or MANOVA with an appropriate post hoc all pairwise multiple comparison test was used to determine significance as indicated in the figure legends. The datasets were considered significantly different if the $P$ value was less than 0.05 (Appendix Table S2). All data were presented as mean ± standard error (SE).

## Data availability

This study includes no data deposited in external repositories.

The source data of this paper are collected in the following database record: biostudies:S-SCDT-10_1038-S44321-024-00103-4.

## Peer review information

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

## Acknowledgements

The authors thank Mr. Isaac S. Demaree for proofreading the manuscript. This work was supported by the National Natural Science Foundation of China, grant numbers 81900402 and 81773631. AGO was supported in part by the NIH grant NS102415.

## Author contributions

**Xingjuan Chen**: Conceptualization; Resources; Data curation; Formal analysis; Supervision; Funding acquisition; Investigation; Writing—original draft; Writing—review and editing. **Yunzheng Yan**: Data curation; Validation; Investigation; Writing—review and editing. **Zhiqiang Liu**: Data curation; Formal analysis; Investigation; Methodology. **Shaokang Yang**: Data curation; Formal analysis; Validation; Investigation; Methodology; Writing—review and editing. **Wei Li**: Data curation; Methodology. **Zhuang Wang**: Data curation; Validation; Methodology; Writing—original draft. **Mengyuan Wang**: Data curation; Validation. **Juan Guo**: Data curation; Validation. **Zhenyang Li**: Validation; Methodology. **Weiyan Zhu**: Validation. **Jingjing Yang**: Data curation; Methodology. **Jiye yin**: Data curation; Methodology. **Qingsong Dai**: Data curation; Methodology. **Yuexiang Li**: Data curation; Methodology. **Cui Wang**: Data curation; Investigation. **Lei Zhao**: Data curation; Methodology. **Xiaotong Yang**: Data curation; Methodology. **Xiaojia Guo**: Data curation; Methodology. **Ling Leng**: Data curation; Investigation. **Jiaxi Xu**: Formal analysis. **Alexander G Obukhov**: Conceptualization; Formal analysis; Supervision; Investigation; Writing—original draft; Writing—review and editing. **Ruiyuan Cao**: Conceptualization; Formal analysis; Supervision; Funding acquisition; Investigation; Writing—original draft; Writing—review and editing. **Wu Zhong**: Conceptualization; Formal analysis; Supervision; Funding acquisition; Writing—original draft; Project administration; Writing—review and editing.

Source data underlying figure panels in this paper may have individual authorship assigned. Where available, figure panel/source data authorship is listed in the following database record: biostudies:S-SCDT-10_1038-S44321-024-00103-4.

## Disclosure and competing interests statement

The authors declare no competing interests.

# Expanded View Figures

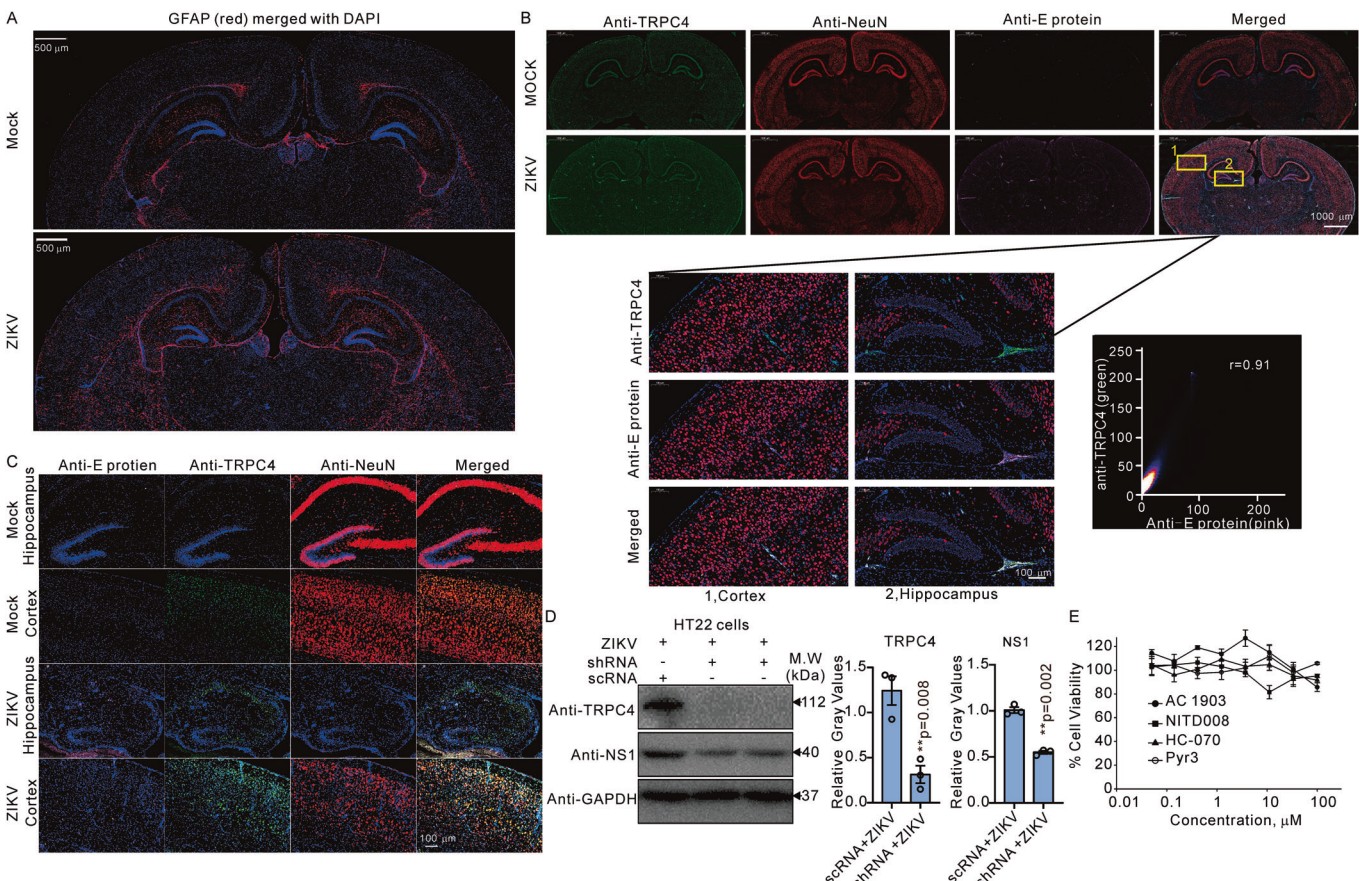

**Figure EV1. The effects of ZIKV infection on brain inflammation and TRPC4 protein levels in A129 mice.**

(A) ZIKV infection results in brain inflammation in A129 mice. Representative images of brain section immunolabeled for GFAP (red) and DAPI (blue). The scale bar represents 500 μm. (B) Representative images of the hippocampus and cortex from mock or ZIKV-infected A129 adult mice at 12 dpi immunolabeled for TRPC4 (green) and viral E protein (red). The scale bars represent 1000 μm and 100 μm, respectively. The right insert in (B) presents the Pearson correlation analysis, revealing an association between TRPC4 protein levels and viral E-protein levels ($r = 0.91$). (C) ZIKV infection also leads to an increase of the TRPC4 protein in neonatal mouse brain. The scale bar represents 100 μm. (D) the knockdown of TRPC4 by siRNA resulted in a reduction in NS1 production in HT22 cells ($n = 3$ biological replicates). (E) the mock-infected cell viability was assessed in the presence of either AC1903, NITD008, HC-070, or Pyr3 ($n = 3$ biological replicates). The unpaired $T$ test (two-tailed) was employed to determine if there was a significant difference between two groups. Data information: In (D, E), data are presented as mean ± SEM, **$P ≤ 0.001$.

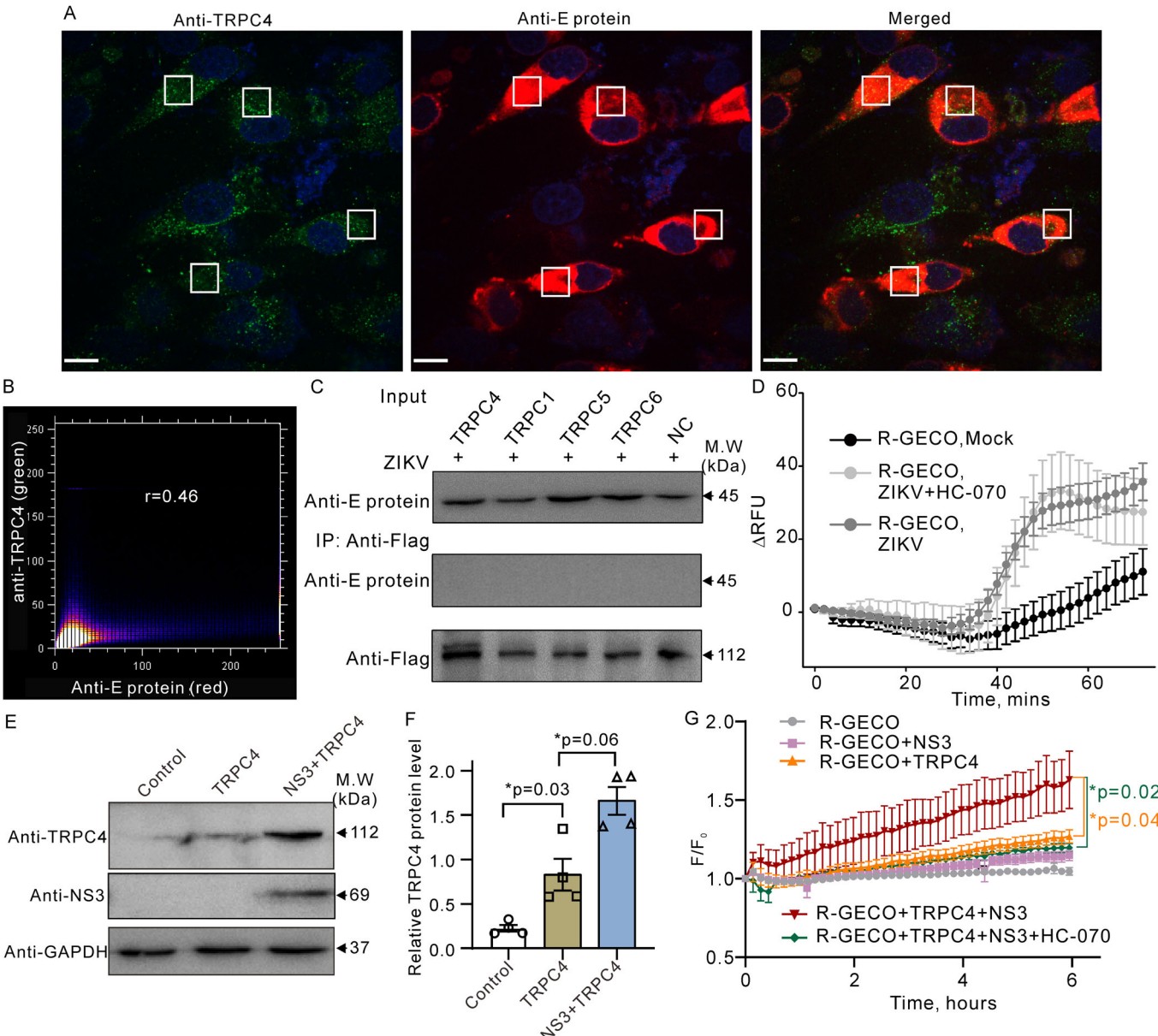

**Figure EV2. The effect of ZIKV proteins on TRPC4 expression and function in BHK cells.**

(**A, B**) BHK cells infected with ZIKV were co-stained with anti-TRPC4 (green) and anti-ZIKV-E-protein (red) antibodies. The scale bar represents 10 μm. The regions of interest (ROI), where correlation analyses are conducted, are depicted in the white boxes. Pearson's Coefficient ($r = 0.46$) was determined to assess the co-localization of the TRPC4 protein and the viral E protein. (**C**) the Co-IP of TRPCs protein and ZIKV-E protein in BHK cells. The anti-flag antibody was used to immunoprecipitate TRPC proteins tagged with flag from cell lysates. The immunoprecipitated complexes were subsequently analyzed using Western blots and probed using the anti-E-protein antibody. The data indicates that there was no association between TRPC and ZIKV-E proteins under our experimental conditions. (**D**) BHK cells were transfected with R-GECO $Ca^{2+}$ sensor plasmid to monitor the intracellular $Ca^{2+}$ levels in mock or ZIKV-infected cells ($n = 4$–5 biological replicates). ΔRFU represents the change in fluorescence intensity relative to the baseline. (**E, F**) A representative immunoblot and the results of densitometry analyses comparing relative TRPC4 protein levels in control, ZIKV-NS3, or ZIKV-NS3 + TRPC4 expressing BHK cells. GAPDH was used as a loading control. Co-expression of ZIKV-NS3 and TRPC4 cDNAs in BHK cells augmented the expression rate of the TRPC4 protein. (**G**) Normalized fluorescence intensity changes (F/F₀) in R-GECO cells co-expressing NS3 and/or TRPC4 in the presence or absence of 10 μM HC-070 ($n = 9$–12; 3–4 wells per each experiment; 3 biological replicates). The MANOVA test with Bonferroni correction was employed to determine if there was a significant difference among multiple groups. Data information: In (**D, F, G**), data are presented as mean ± SEM.

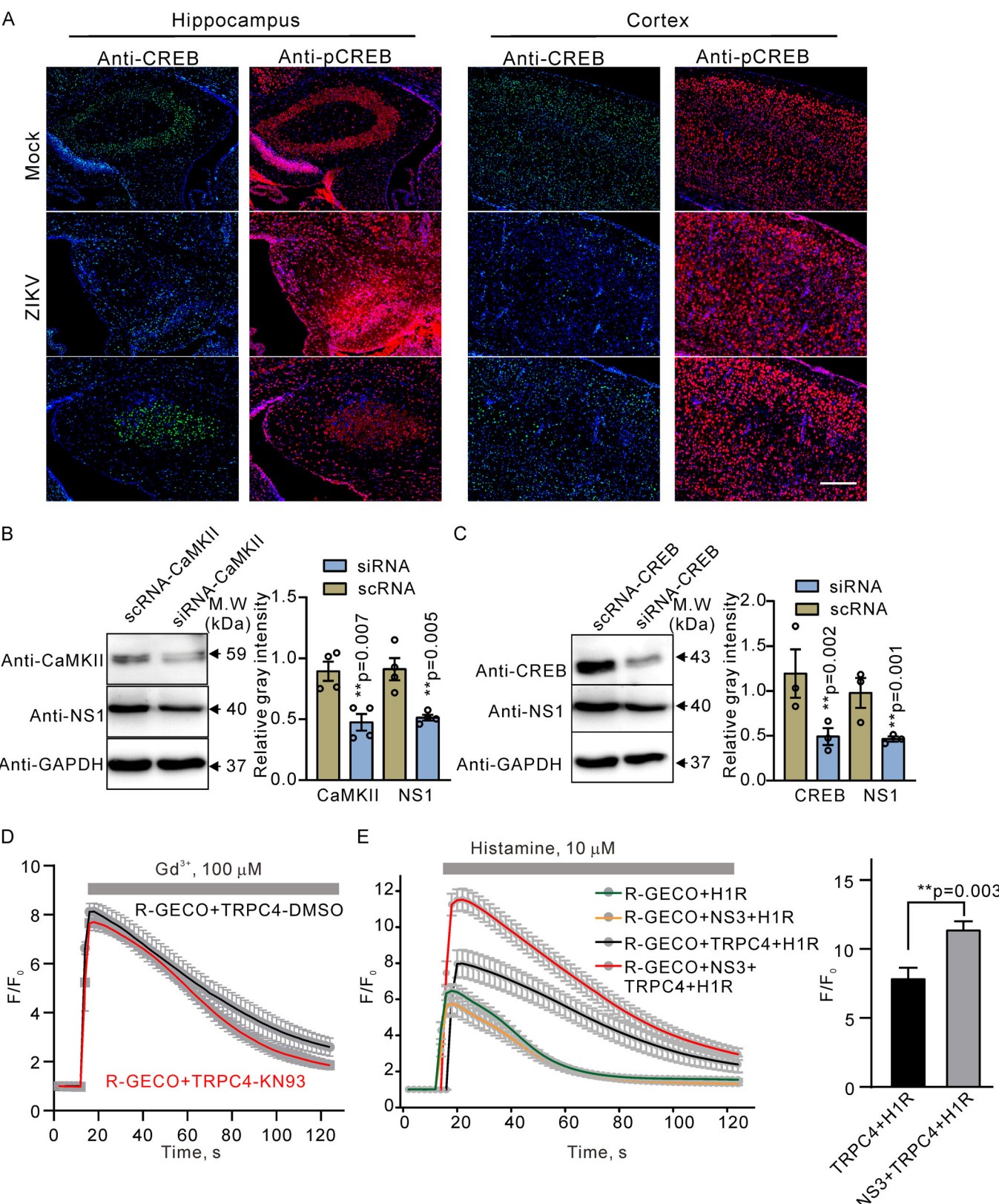

**Figure EV3. ZIKV-NS3 protein enhances TRPC4-mediated calcium signaling via the CaMKII-CREB pathway in neuronal cells.**

(A) ZIKV infection of the neonatal mouse brain elevates the protein level of pCREB. 1-day-old ICR mice were infected with ZIKV, and then the brains were collected at 12 dpi. Tissues were fixed and stained with anti-CREB (green), and anti-pCREB (red) antibodies. The scale bar represents 100 μm. (B, C) Representative Western blot images demonstrate the effectiveness of siRNA-induced reduction of ZIKV-NS1 protein production in HT22 cells ($n = 4$ biological replicates). Prior to exposing the cells to ZIKV (MOI 0.01), a 48-h transfection was performed using scRNAs or siRNAs that specifically target CaMKII ($n = 4$ biological replicates) or CREB ($n = 3$ biological replicates). (D) cells were transfected with R-GECO $Ca^{2+}$ sensor plasmid to monitor the intracellular $Ca^{2+}$ levels. No difference was observed between the $Gd^{3+}$ (100 μM)-induced fluorescence increases in the presence of the vehicle (DMSO) or 10 μM KN-93 in R-GECO + TRPC4 expressing cells ($n = 6$ biological replicates). $Gd^{3+}$ was applied at the times indicated with the horizontal bar. (E) The averaged normalized fluorescence increases induced by histamine (10 μM) in H1R (histamine receptor, green line), H1R + NS3 (orange line), TRPC4 + H1R (black line), or NS3 + TRPC4 + H1R (red line) expressing cells are shown (n = 6 biological replicates). Histamine was added at the times indicated by the horizontal bar. (E) Right panel, a comparison of averaged peak values for data shown in the left panel. The unpaired *T* test (two-tailed) was employed to determine if there was a significant difference between two groups. Data information: In (B–E), data are presented as mean ± SEM, **$P \leq 0.001$.

