## [Peer Review File · EMBO Molecular Medicine]

In vitro and in vivo inhibition of the host TRPC4 channel attenuates Zika virus infection

Xingjuan Chen, Yunzheng Yan, Zhiqiang Liu, Shaokang Yang, Wei Li, Zhuang Wang, Mengyuan Wang, Juan Guo, Zhenyang Li, Weiyang Zhu, Jingjing Yang, Jiye Yin, Qingsong Dai, Yuexiang Li, Cui Wang, Lei Zhao, Xiaotong Yang, Xiaojia Guo, Ling Leng, Jiayi Xu, Alexander Obukhov, Ruiyuan Cao, and Wu Zhong

Corresponding authors: Alexander Obukhov (aobukhov@iu.edu), Wu Zhong (zhongwu@bmi.ac.cn), Ruiyuan Cao (caoruiyuan@bmi.ac.cn)

Review Timeline:

Submission Date:	12th Sep 23
Editorial Decision:	6th Oct 23
Revision Received:	14th Feb 24
Editorial Decision:	6th Mar 24
Revision Received:	4th Jun 24
Editorial Decision:	18th Jun 24
Revision Received:	24th Jun 24
Accepted:	1st Jul 24

Editor: Zeljko Durdevic

Transaction Report:

6th Oct 2023

Dear Dr. Chen,

Thank you for the submission of your manuscript to EMBO Molecular Medicine. We have now received feedback from the three reviewers who agreed to evaluate your manuscript. While the referee #1 is overall supportive, referee #3 recognizes interest of the study but raises number of important concerns and referee #2, while acknowledging interest of the study, recommends rejection of the manuscript due to the lack of experimental evidence to support proposed mechanism. Our cross-commenting session made clear that elucidation of the mechanism is essential for the further consideration of the manuscript. I would also recommend running the article by a native English speaker. If you would like to discuss further the points raised by the referees, I am available to do so via email or video. Let me know if you are interested in this option.

We would welcome the submission of a revised version within three months for further consideration. Please let us know if you require longer to complete the revision.

I look forward to receiving your revised manuscript.

Yours sincerely,

Zeljko Durdevic

We require:

- 1) A .docx formatted version of the manuscript text (including legends for main figures, EV figures and tables). Please make sure that the changes are highlighted to be clearly visible.
- 2) Individual production quality figure files as .eps, .tif, .jpg (one file per figure). For guidance, download the 'Figure Guide PDF': (<https://www.embopress.org/page/journal/17574684/authorguide#figureformat>).
- 3) A .docx formatted letter INCLUDING the reviewers' reports and your detailed point-by-point responses to their comments. As part of the EMBO Press transparent editorial process, the point-by-point response is part of the Review Process File (RPF), which will be published alongside your paper.
- 4) A complete author checklist, which you can download from our author guidelines (<https://www.embopress.org/page/journal/17574684/authorguide#submissionofrevisions>). Please insert information in the checklist that is also reflected in the manuscript. The completed author checklist will also be part of the RPF.
- 5) Please note that all corresponding authors are required to supply an ORCID ID for their name upon submission of a revised manuscript.

6) It is mandatory to include a 'Data Availability' section after the Materials and Methods. Before submitting your revision, primary datasets produced in this study need to be deposited in an appropriate public database, and the accession numbers and database listed under 'Data Availability'. Please remember to provide a reviewer password if the datasets are not yet public (see <https://www.embopress.org/page/journal/17574684/authorguide#dataavailability>).

.

- the medical issue you are addressing,

- the results obtained and

- their clinical impact.

13) Author contributions: You will be asked to provide CRediT (Contributor Role Taxonomy) terms in the submission system. These replace a narrative author contribution section in the manuscript.

14) A Conflict of Interest statement should be provided in the main text.

Please note: When submitting your revision you will be prompted to enter your funding and payment information. This will allow Wiley to send you a quote for the article processing charge (APC) in case of acceptance. This quote takes into account any reduction or fee waivers that you may be eligible for. Authors do not need to pay any fees before their manuscript is accepted and transferred to the publisher.

EMBO Press participates in many Publish and Read agreements that allow authors to publish Open Access with reduced/no publication charges. Check your eligibility: <https://authorservices.wiley.com/author-resources/Journal-Authors/open-access/affiliation-policies-payments/index.html>

**** Reviewer's comments ****

Referee #1 (Remarks for Author):

The Zika virus is a significant pathogen with the potential for global pandemic, and research on anti-ZIKV measures holds great significance due to the absence of available antiviral drugs or vaccines. In this work, the authors provided the first evidence supporting the role of TRPC4 in ZIKV infection, which was found to be upregulated through the CaMK II-CREB signaling pathway upon invasion by ZIKV. Notably, manipulation of host TRPC4 by ZIKV may contribute to a range of neuropathological consequences. Furthermore, inhibition of both TRPC4 and CaMK II effectively mitigated infection pathogenesis, symptoms, and mortality in mice. Additionally, the authors identified HC-070 as a small molecular inhibitor that alleviated ZIKV infection, further supporting TRPC4 as a promising target for novel antiviral therapy.

Overall, the topic and initial discoveries appear interesting. In general, most of work has been carefully done, with sound research methodologies and significant contributions to the field of anti-ZIKV agents. Here are some questions and suggestions that authors may find useful for improving this manuscript:

- 1) HC-070 and ML204 are commonly used TRPC4/5 antagonists with IC50s of 46 nM and 0.96 μ M, respectively, while HC-070 and ML204 showed anti-ZIKV activity with an IC50 value of 4.5 {plus minus} 3.0 μ M and 58.0{plus minus}4.5 μ M, approximately 50-100 fold higher. The authors shall comment on this discrepancy.
- 2) Interestingly, in time of addition experiment NITD008 effectively suppressed ZIKV RNA production in stages V, while HC-070 significantly lost antiviral activity. The authors shall comment on this discrepancy.
- 3) The authors proposed that ZIKV infection triggers intracellular Ca²⁺ increases to promote TRPC4 expression via the activation of CaMKII-CREB, using a set of small molecule inhibitors. Given the potential off-target effect of these inhibitors, the lost of function study using KO or KD of interesting gene would provide more convincing evidence to enhance the conclusion.
- 4) The Anti-FLAV protein in Figure 1A should be consistent with the Anti-E protein in Figure 3G.
- 5) In Figure 1B, FLAV of Y axis is quite confusing, why not use ZIKV?
- 6) Please confirm that the "+" and "-" in Figure 2B are correct.
- 7) Please explain why Figure 3D used NS5 protein, while Figure 3E and F used NS1 protein.
- 8) Please confirm that the "+" and "-" in Figure 3F are correct.
- 9) In Figure 1C, the molecular weight of α -tubulin should be 55kd; in Figure 1G, the MW of GADPH should be 37kd.

Referee #2 (Comments on Novelty/Model System for Author):

The identification of TRPC4 as a key host factor in ZIKV infection by this paper is appreciated. The drug discovery of a TRPC4 inhibitor as a potential therapeutics for ZIKV infection is also significant. However, the molecular mechanisms by which ZIKAV induces TRPC4 expression or activates its channel activity, and TRPC4 promotes ZIKV infection still need to be made clear because no data supporting the authors' claims have been provided in the manuscript. Thus, this manuscript should be rejected.

Referee #2 (Remarks for Author):

Manuscript #: EMM-2023-18685-T

In this manuscript, Chen and colleagues investigated the involvement of the TRPC4 cation channel in Zika virus (ZIKV) infection. The authors demonstrated that ZIKV infection increased TRPC4 expression, and knockdown of TRPC4 or pharmacological inhibition of TRPC4 attenuated ZIKV infection *in vitro*. They further showed that treatment with TRPC4 inhibitor prolonged the survival of ZIKV-infected mice.

It is appreciated the identification of TRPC4 as a key host factor in ZIKV infection; however, this study is too immature to support the authors' conclusion. Specifically, it is unclear how broadly significant the findings are without understanding underlying molecular mechanisms. Thus, this reviewer deems the manuscript more suitable for a specialized journal.

Major concerns:

1 The role of TRPC4 in ZIKV infection is unclear: In the Abstract, the authors claim that increased TRPC4 expression during ZIKV infection promotes virus entry. However, no data have been provided to support this. Instead, TRPC4 does not contribute to increased intracellular calcium concentration during ZIKV infection (Figure EV3C), indicating that this channel is dispensable for virus entry. In addition, the TRPC4 function that regulates ZIKV infection should also be evaluated.

Alternatively, it was shown that treatment with TRPC4 inhibitors in the late stage of the infection process is more efficient in suppressing the infection than that in the early stage in Figures 4A and 4B, suggesting that TRPC4 inhibition suppresses the processes later than viral entry, including replication.

2 A molecular mechanism through which ZIKAV infection upregulates TRPC4 expression is controversial: The authors speculated that ZIKV transcriptionally upregulates TRPC4 expression via the CaMKII-CREB pathway (Page 8, lanes 256-258); however, CREB phosphorylation was not promoted by ZIKV infection (lanes 1 and 2 in Figures 4F), although quantitative data on the band intensities of multiple immunoblots are missing. Therefore, the molecular mechanism by which ZIKV promotes TRPC4 expression (via CaMKII) is yet to be determined. In this line, it has been reported that CaMKII regulates TRPC activity by binding to TRPC (Vinayagam et al., *eLife* 9: e60603, 2020). The authors must cite such mandatory papers and test whether CaMKII directly regulates TRPC4 activity or activates its gene expression via transcription factors other than CREB. These experiments may help explain why the inhibitory effect of TRPC4 inhibitor on virus infection is cell context-dependent [HC-070 treatment did not reduce the copy number of ZIKV RNA in Vero, Hhu7, and A127 cells (Page 6, lines 153-154)]. However, the CaMKII-CREB pathway works ubiquitously in a variety of cell types.

General comments:

1 This manuscript includes many typos/errors and insufficient/inadequate descriptions. Therefore, it might be difficult for readers to understand the experiment performed and the results derived from the corresponding experiments or to reproduce the experiments.

1.1 More information will be needed:

- Figure numbers in the figure files. Due to the lack of this, linking the corresponding figures and the text was difficult. The authors should include such information for further review processes in the future.
- More information on each experiment performed in the Figure legends (albeit briefly). Although it is described in the Material and Methods, a brief note in the Figure legends helps readers understand how the figures were obtained. For example, it is unclear what are "Summary data" in lines 624, 651, 655, and 672 on pages 19-20.
- Page 19, line 646: info for cells used.
- Figure 1A: an explanation for TRPC4-BHK.
- Figure 3D: a rational reason why the expression level of NS5, but not NS1, was quantitated.
- Figure 6C: error bars on the graph.
- Figure EV2B: a label on the vertical axis in the graph in "insert."

1.2 The following might be typos:

- Page 5, line 136: Fig 2C might be Fig 2D.
- Page 5, line 138: Fig 2D might be Fig 2C.
- Page 7, line 191: Fig EV 2C might be Fig EV 3C.
- Page 7, line 196: Fig 4A might be Fig 4C.
- Page 8, line 249: Fig 6D might be Fig 6E.
- Figure 3A: HC-007 should be HC-070.

2 Appropriate negative controls are required in Figure 1A (mock-infected cells) and Figure 3A (untreated cells).

3 Please consult an expert in statistics to ensure the statistical tests' validity. For example, two-way ANOVA tests the hypothesis that there is no difference among three or more groups but does not demonstrate the difference between specific two groups. Multiple comparisons must follow two-way ANOVA once it detects differences (Figure 5C, D).

Minor comments:

1 Figure 1G: The authors claim that TRPC4 mRNA and viral RNA levels are highly correlated in Figure 1H; however, such correlation is no longer observed at a protein level (Figure 1G). Specifically, TRPC4 expression levels are almost constant among lanes 2-4, whereas the amount of NS1 proteins is dramatically upregulated in lane 4. An explanation for this discrepancy is needed.

2 An appropriate explanation of why E protein and NS1 were used differently in each experiment in quantifying viral infection is needed.

3 Figure EV3A: Saturated fluorescence signals may not accurately evaluate the colocalization. Please consult with an expert in

image analysis.

4 Figure 6: Authors should quantify the virus replication.

Referee #3 (Comments on Novelty/Model System for Author):

This study employed a comprehensive approach using *in vitro* BHK21 cells, *ex vivo* human brain organoids, and *in vivo* mouse models to assess the functions of the proposed chemicals. Expanding the investigation to additional neuron cell types would further enhance the findings.

Referee #3 (Remarks for Author):

Zika virus (ZIKV) has been rampant in South America and persistently endemic in regions of Southeast Asia, Africa, and South America in recent years. Infections with ZIKV have been linked to various neurological outcomes, particularly microcephaly in newborns. Dr. Chen and his/her team delved into the neurological manifestations associated with ZIKV infection, pinpointing TRPC4, a calcium channel distinctly implicated in ZIKV infection and its subsequent outcomes. Following studies revealed that ZIKV infection increase the TRPC4 expression via CaMKII-CREB pathway. Notably, antagonists to TRPC4 like HC-070 and the CaMKII inhibitor KN93 effectively curtailed ZIKV proliferation and mitigated related neurological symptoms in *in vitro* (BHK21 cells), *ex vivo* (Human brain organoids), and *in vivo* (ICR suckling mice) models. This study comprehensively evaluated the efficacy of HC-070 and KN93 across various models, presenting potential antiviral therapeutic options for neurological complications linked to ZIKV.

The manuscript was logically organized and well written, but the qualities of some experiments still need to be improved to solidify the conclusion well. Specific recommendations are as follows:

1. Fig. 1A, the image resolution does not sufficiently facilitate the determination of whether there's a colocalization between elevated TRPC4 expression and ZIKV proteins within cells. In certain ZIKV-infected cells, an apparent increase in TRPC4 expression is not discernible. Moreover, several cells manifesting elevated TRPC4 levels did not appear to be infected by ZIKV. It would be beneficial to provide some images at a higher magnification, which could be incorporated into the Supplementary data. Furthermore, it is advisable to include the MOCK group to showcase the basal expression level of TRPC4 in cells.
2. Fig. 1B, 1D, and corresponding sub-figures, the methodology for calculating grey values remains unclear. Is it based on the fluorescence intensity of all cells captured in the images? This has not been explicitly delineated either in the methods section or the figure legends. Such a quantification technique may not sufficiently capture the evaluate of ZIKV infection or TRPC4 expression changes. Employing grey intensity calculations from Western Blot images (semi-quantification) might offer a more dependable quantification compared to fluorescence evaluation. Alternatively, quantifying the number of infected cells could serve as an alternative measure.
3. Fig. 1B, the pooling of data points from both the TRPC4 over-expressed group and the normal infection group for correlation analysis seems inappropriate. It would be more appropriate to analyze these groups separately, or alternatively, consider excluding the TRPC4 over-expressed group from the analysis.
4. For a deeper understanding of the relationship between TRPC4 and ZIKV infection, exploring varying levels of TRPC4 overexpression in a dose-dependent manner during ZIKV infection could offer valuable insights.
5. There's a noticeable discrepancy between Fig. 1C and Fig. 1D. While TRPC4 expression level in Fig. 1C shows a 1.6-fold change, the variance is considerably greater in the Western Blot (Fig 1D). This inconsistency raises questions about the accuracy of the fluorescence determination.
6. Fig. 1E, the right panel, comparing the ZIKV RNA levels between the MOCK (un-infected) group and the ZIKV-infected group seems almost redundant or without significant insight.
7. For ZIKV detection, the authors employed a variety of antibodies to identify ZIKV infection, encompassing ZIKV NS1, E, and NS5 proteins. What was the rationale behind selecting different detection targets within a series of similar experiments? For TRPC4 detection, is there a possibility that the antibodies employed to detect TRPC4 may cross-react with other proteins from the TRPC family?
8. There appears to be a discrepancy between the figure labels for Fig 2C and 2D, as they seem to be reversed in contrast to the descriptions provided in the main manuscript.
9. Given the sensitivity of ZIKV particles to certain chemicals, particularly shifts in pH, was there a preliminary assessment to ascertain if HC-070 and KN93 could directly neutralize ZIKV upon co-incubation?
10. Fig. 4B, for the column termed as "virus", does it mean the non-treated group?

11. For ZIKV, viral particle entry spans from the initial time point up to 12 hours. The onset of transcription typically starts around 1-2 hours post-entry, followed by the replication process. As deduced from Fig. B, HC-070 appears to have a potent effect during stages III and IV, a moderate impact in stage II. This suggests that the compound primarily functions between the 2hr to 4hr, predominantly affecting the transcription and partially the replication processes. HC-070 showed no co-localization or binding activity to the viral particle, suggesting it has minimal influence on viral entry. Also, it likely has no impact on virus assembly and shows no influence during stage V. Hence, the manuscript's conclusion that "HC-070 acted at almost the entire viral life cycle" doesn't seem precise.

12. The precise mechanism by which HC-070 and the calcium-related pathway take their effects remains to be elucidated. While only TRPC4 is implicated and not other similar TRPC family proteins-despite their relation to calcium regulation-it suggests that TRPC4 might directly interact with certain viral proteins within cells. In vivo Co-IP experiments could be employed to shed light on this aspect.

13. Line 191, the reference to Fig. EV 2C seems incorrect. Additionally, Line 196, the reference should likely be to Fig. 4C instead of Fig. 4A.

14. Interestingly, the extracellular EGTA treatment suppressed viral production, implying that the viral life cycle depends on calcium influx from the outside. However, this observation might not be directly related to HC-070, as its primary effect is intracellular.

15. Fig 4F, pCREB appears more prominently detected by antibodies compared to regular CREB. Could this discrepancy be attributed to differences in binding efficiency during Western blot detection?

16. Fig. 6E, all mice in the control group were dead by day 15. How were their Racine scores measured for the behavioral experiments on days 15 and 18 post-infection?

17. Though TRPC4 inhibition has been identified to mitigate seizure symptoms, understanding its correlation with suppressing ZIKV production remains intriguing. Is the reduction of ZIKV integral to the seizure symptom relief, or are these two separate narratives?

18. The ethics statement should be provided, especially given the use of animals and human tissues in the study.

Responses to Editor

We thank the Editor and Reviewers for valuable comments that helped us improve the manuscript.

Response to the Editor

We thank the Editor for allowing us to revise our manuscript.

Editor: "Our cross-commenting session made clear that elucidation of the mechanism is essential for the further consideration of the manuscript."

Response: Our revised manuscript is now focused on elucidating the mechanisms.

Editor: "I would also recommend running the article by a native English speaker."

Response: The revised manuscript was edited by a native English speaker.

Response to the Reviewers

Referee #1: "...Overall, the topic and initial discoveries appear interesting. In general, most of work has been carefully done, with sound research methodologies and significant contributions to the field of anti-ZIKV agents."

Response: Thank you for your positive evaluation of our manuscript and valuable comments which helped us significantly improve our manuscript.

Referee #1: “HC-070 and ML204 are commonly used TRPC4/5 antagonists with IC₅₀s of 46 nM and 0.96 μM, respectively, while HC-070 and ML204 showed anti-ZIKV activity with an IC₅₀ value of 4.5 {plus minus} 3.0 μM and 58.0{plus minus}4.5 μM, approximately 50-100 fold higher. The authors shall comment on this discrepancy.”

Response: Indeed, HC-070 and ML204 exhibit dose-dependent inhibition of calcium influx induced by Gd³⁺ and GPCR agonist-activated TRPC4 channels in HEK cells overexpressing TRPC4 channels with lower IC₅₀s than that of ML204. The sentence was revised in the manuscript to provide a clearer explanation. Here, in order to assess the antiviral activity of these compounds, we measured the survival rate of infected cells treated with them. During this process, multiple cellular signaling pathway mechanisms may be influenced by channel-mediated calcium influx may contribute to the discrepancy in determining the IC₅₀ value. Nevertheless, it is evident that HC070 with more potent inhibition for channel-mediated calcium influx, also demonstrates a stronger antiviral activity.

Referee #1: “Interestingly, in time of addition experiment NITD008 effectively suppressed ZIKV RNA production in stages V, while HC-070 significantly lost antiviral activity. The authors shall comment on this discrepancy.”

Response: In the time of addition experiment, HC-070 seemed to be effective both in the attachment stage and in the replication stage, which suggested that the drug molecule might interact with viral particles, or that the viral E protein might interact with host TRPC4 on the cell surface. However, the results obtained from the temperature-dependent infectivity inhibition assay indicated that HC-070 could not affect the infectivity of viral particles (revised Fig 5C). Further, the results from Co-IP assay suggested the interaction between viral E protein and TRPC4 was negligible (revised Fig EV 3C). Thus, it is highly probable that HC-070 exerts its action during the replication phase of ZIKV. Literature implicates a vital role for the organellar Ca²⁺ dynamics in regulating virus entry, replication, and severity of the infection (reviewed by PMID: 34304899). In stage V, the statistical analysis revealed that there was decreased inhibition of viral replication by HC-070, which indicates essential role of calcium influx mediated by TRPC4 for the whole stage of replication. We further discussed on this result in the revised manuscript.

Referee #1: “The authors proposed that ZIKV infection triggers intracellular Ca²⁺ increases to promote TRPC4 expression via the activation of CaMKII-CREB, using a set of small molecule inhibitors. Given the potential off-target effect of these inhibitors, the lost of function study using KO or KD of interesting gene would provide more convincing evidence to enhance the conclusion.”

Response: We greatly appreciate your valuable suggestion. To down-regulate CAMKII levels, we employed gene knockdown by siRNA, which effectively suppressed NS1

production to a comparable extent as observed with KN93 treatment. The corresponding results are illustrated in Figure 4I.

Referee #1: “The Anti-FLAV protein in Figure 1A should be consistent with the Anti-E protein in Figure 3G.”

Response: This was an oversight on our part. We have changed “anti-FLAV” into anti-E protein in Figure 1.

Referee #1: “In Figure 1B, FLAV of Y axis is quite confusing, why not use ZIKV?”

Response: We apologize for the inconsistent naming of the flavivirus E protein in the manuscript and appreciate the reviewer's comments. In this experiment, we utilized monoclonal anti-flavivirus group antigen antibody (clone D1-4G2-4-15, protein E). Therefore, we consistently referred to this protein as 'E protein' in all immunofluorescence experiments described in the revised manuscript.

Referee #1: “Please confirm that the “+” and “-” in Figure 2B are correct.”

Response: Thank you for noticing this error. We corrected it in the revised manuscript.

Referee #1: “Please explain why Figure 3D used NS5 protein, while Figure 3E and F used NS1 protein.”

Response: We apologize for inconsistent usage of both ZIKV-NS5 and NS1 antibodies in figure 3. The application of HC-070 or KN93 inhibits the expression of viral proteins,

including NS1 and NS5. To ensure consistency throughout the article, we conducted additional experiments using an NS1 antibody instead of an NS5 antibody and replaced the images in Figure 3D. Consequently, the revised manuscript incorporates WB data utilizing NS1 antibody.

Referee #1: "Please confirm that the "+" and "-" in Figure 3F are correct"

Response: This was an oversight on our part. The symbol has been corrected.

Referee #1: "In Figure 1C, the molecular weight of α -tubulin should be 55kd; in Figure 1G, the MW of GADPH should be 37 kd."

Response: We apologize for this confusion. The arrow in Figure 1 indicated the protein marker band closest to the target band in the raw WB image. To eliminate this confusion, we have revised the placement of the arrow to indicate the molecular weight of the target band.

Referee #2: "The identification of TRPC4 as a key host factor in ZIKV infection by this paper is appreciated. The drug discovery of a TRPC4 inhibitor as a potential therapeutics for ZIKV infection is also significant."

Response: We thank this reviewer for these encouraging comments.

Referee #2: "However, the molecular mechanisms by which ZIKAV induces TRPC4 expression or activates its channel activity, and TRPC4 promotes ZIKV infection still

need to be made clear because no data supporting the authors' claims have been provided in the manuscript.

Response: We thank this reviewer for important suggestions. Our revised manuscript focuses on the molecular mechanisms.

Referee #2: "It is appreciated the identification of TRPC4 as a key host factor in ZIKV infection; however, this study is too immature to support the authors' conclusion."

Response: We performed additional experiments and obtained new data to further support our conclusions. The new data are now included in the revised manuscript.

Referee #2: "Specifically, it is unclear how broadly significant the findings are without understanding underlying molecular mechanisms."

Response: We performed additional experiments to better understand underlying molecular mechanisms.

Referee #2: "The role of TRPC4 in ZIKV infection is unclear: In the Abstract, the authors claim that increased TRPC4 expression during ZIKV infection promotes virus entry. However, no data have been provided to support this. Instead, TRPC4 does not contribute to increased intracellular calcium concentration during ZIKV infection (Figure EV3C), indicating that this channel is dispensable for virus entry."

Response: We apologize for any confusion caused by these sentences in the Abstract and have rephrased them. The role of TRPC4 in viral infection was further elucidated

through additional experiments in the revised manuscript, including Co-IP of TRPC4 with viral E proteins, infectivity assays, DDX3X assays, and calcium imaging experiments.

Referee #2: “In addition, the TRPC4 function that regulates ZIKV infection should also be evaluated.”

Response: The function of TRPC4 channel was tested using live cell calcium imaging in cell overexpressed ZIKV-NS3 and TRPC4 channels. As shown in Fig 4M and EV4E, the ZIKV-NS3 protein enhanced the calcium influx mediated by TRPC4 channel.

Referee #2: “Alternatively, it was shown that treatment with TRPC4 inhibitors in the late stage of the infection process is more efficient in suppressing the infection than that in the early stage in Figures 4A and 4B, suggesting that TRPC4 inhibition suppresses the processes later than viral entry, including replication.”

Response: We totally agree that the TRPC4 inhibitor has less effect the early stage of viral life cycle. To address this issue, the substantial interaction between viral E protein and TRPC4 was measured using the IP assay and hardly any interaction was observed (revised Fig EV 3C). These results suggest that the antiviral activity of TRPC4 inhibitors may not target the initial phase of viral infection. It has been previously reported that host DDX3X plays a crucial role in ZIKV replication (PMID: 32787106), and our results also revealed that overexpression of TRPC4 can enhance nuclear localization of DDX3X, indicating potential involvement of TRPC4 or channel-mediated calcium signaling in regulating ZIKV replication. We further discussed this issue in the revised manuscript.

Referee #2: “A molecular mechanism through which ZIKAV infection upregulates TRPC4 expression is controversial: The authors speculated that ZIKV transcriptionally upregulates TRPC4 expression via the CaMKII-CREB pathway (Page 8, lanes 256-258); however, CREB phosphorylation was not promoted by ZIKV infection (lanes 1 and 2 in Figures 4F), although quantitative data on the band intensities of multiple immunoblots are missing.”

Response: Thank you for these important comments. The quantification of data has been done and included in the revised manuscript. Indeed, ZIKV infection slightly increased the levels of CREB in host cells; however, no significant statistical difference was observed. Subsequently, we employed the siRNA technique to knock down CaMKII or CREB levels, resulting in a significant reduction in viral protein production (revised Fig 4I-L). This exemplifies the viral reliance on host cell CaMKII and CREB.

Referee #2: “Therefore, the molecular mechanism by which ZIKV promotes TRPC4 expression (via CaMKII) is yet to be determined. In this line, it has been reported that CaMKII regulates TRPC activity by binding to TRPC (Vinayagam et al., eLife 9: e60603, 2020). The authors must cite such mandatory papers and test whether CaMKII directly regulates TRPC4 activity or activates its gene expression via transcription factors other than CREB. These experiments may help explain why the inhibitory effect of TRPC4 inhibitor on virus infection is cell context-dependent [HC-070 treatment did not reduce the

copy number of ZIKV RNA in Vero, Hhu7, and A127 cells (Page 6, lines 153-154)].

However, the CaMKII-CREB pathway works ubiquitously in a variety of cell types.”

Response: Thank you for this point. The paper of Vinayagam describes interaction of calmodulin and TRPC4. We have cited this article in the revised manuscript. To verify the effect of CaMKII on TRPC4 activity, we examined the effect of acute KN93 (CaMKII specific inhibitor) on TRPC4-mediated calcium influx. As depicted in Fig. EV4B, acute use of KN93 did not exert any influence on TRPC4-mediated calcium influx. Consequently, based on our experimental conditions, the direct regulatory role of CaMKII in relation to TRPC4 appears to be negligible.

Referee #2: “This manuscript includes many typos/errors and insufficient/inadequate descriptions. Therefore, it might be difficult for readers to understand the experiment performed and the results derived from the corresponding experiments or to reproduce the experiments.”

Response: We apologize for several typos/errors and inadequate descriptions in the original version of our manuscript. We diligently proofread our manuscript and provided the detailed descriptions of methods we used in this study.

Referee #2: “More information will be needed: Figure numbers in the figure files. Due to the lack of this, linking the corresponding figures and the text was difficult. The authors should include such information for further review processes in the future.”

Response: Figure numbers are now provided in the revised manuscript.

Referee #2: “More information on each experiment performed in the Figure legends (albeit briefly). Although it is described in the Material and Methods, a brief note in the Figure legends helps readers understand how the figures were obtained. For example, it is unclear what are "Summary data" in lines 624, 651, 655, and 672 on pages 19-20.”

Response: We revised figure legends to rectify the problem.

Referee #2: “Page 19, line 646: info for cells used.”

Response: The cell line information has been added in the figure legend.

Referee #2: “Figure 1A: an explanation for TRPC4-BHK.”

Response: TRPC4-BHK stands for “BHK cells overexpressing TRPC4.” This experimental group was removed.

Referee #2: “Figure 3D: a rational reason why the expression level of NS5, but not NS1, was quantitated.”

Response: Thank you for this important comment. To ensure consistency throughout the manuscript, we conducted additional experiments with an NS1 antibody instead of an NS5 antibody and now included a new panel in Figure 3D.

Referee #2: “Figure 6C: error bars on the graph.”

Response: This group of animals had normal activity, so they all had the same highest score (SEM=0).

Referee #2: "Figure EV2B: a label on the vertical axis in the graph in "insert.""

Response: The missing label for the axis was added.

Referee #2: "The following might be typos:

- Page 5, line 136: Fig 2C might be Fig 2D.
- Page 5, line 138: Fig 2D might be Fig 2C.
- Page 7, line 191: Fig EV 2C might be Fig EV 3C.
- Page 7, line 196: Fig 4A might be Fig 4C.
- Page 8, line 249: Fig 6D might be Fig 6E.
- Figure 3A: HC-007 should be HC-070."

Response: We thank the reviewer for noticing these typos. We corrected them in the revised manuscript.

Referee #2: "Appropriate negative controls are required in Figure 1A (mock-infected cells) and Figure 3A (untreated cells)."

Response: We apologize for this oversight. Figure 1A now includes the images from the mock group, while the effects of the compounds used in Figure 3A on uninfected cells (without ZIKV) are shown in Figure EV2E.

Referee #2: “Please consult an expert in statistics to ensure the statistical tests' validity.

For example, two-way ANOVA tests the hypothesis that there is no difference among three or more groups but does not demonstrate the difference between specific two groups. Multiple comparisons must follow two-way ANOVA once it detects differences (Figure 5C, D).”

Response: We sincerely apologize for the inaccuracies in the description of statistical methods and the incorrect usage of some statistical methods in Figure 5. We sought guidance from our colleagues proficient in statistics and reanalyzed our data.

Referee #2: “Figure 1G: The authors claim that TRPC4 mRNA and viral RNA levels are highly correlated in Figure 1H; however, such correlation is no longer observed at a protein level (Figure 1G). Specifically, TRPC4 expression levels are almost constant among lanes 2-4, whereas the amount of NS1 proteins is dramatically upregulated in lane 4. An explanation for this discrepancy is needed.”

Response: Indeed, we observed upregulation of the TRPC4 protein in ZIKV-infected mouse brain tissue. However, the correlation between the TRPC4 protein and the viral protein was weak. The precise pathological mechanism underlying ZIKV infection in both neurons and glial cells remains elusive. Subdividing brain regions and focusing on specific cell types may provide valuable insights into elucidating the intricate relationship between TRPC4 and viral proteins. We discussed the points in the revised manuscript.

Referee #2: “An appropriate explanation of why E protein and NS1 were used differently in each experiment in quantifying viral infection is needed.”

Response: The antibody used in the immunofluorescence experiment is anti-flavivirus group antigen antibody (clone D1-4G2-4-15, protein E), specifically designed for IF. According to the instructions, this antibody is generally not suitable for WB, so we opted for other available ZIKV protein antibodies such as anti-NS1. We have explained this in the revised manuscript's methods and materials section.

Referee #2: “Figure EV3A: Saturated fluorescence signals may not accurately evaluate the colocalization. Please consult with an expert in image analysis.”

Response: We used ImageJ software to analyze the images included in Figure EV3A and determined that all gray values for all fluorescence signals in the images were below 200. Thus, all the values were lower than the saturation threshold of 255 (please see the following picture).

Referee #2: “Figure 6: Authors should quantify the virus replication.”

Response: We now quantified the viral loads in the brain tissues from ICR suckling mice using qRT-PCR. The quantification results are presented in Figure 7C.

Referee #3: “This study employed a comprehensive approach using in vitro BHK21 cells, ex vivo human brain organoids, and in vivo mouse models to assess the functions of the proposed chemicals.”

Response: We thank the reviewer for noting that we used a comprehensive approach.

Referee #3: “Expanding the investigation to additional neuron cell types would further enhance the findings.”

Response: Thank you for this suggestion. We performed additional experiments to address this point. To further investigate the role of the host TRPC4 channel in ZIKV

infection, we utilized HT22 cells (mouse embryonic hippocampal neuronal cell line) and U87 cells (human glioma cell line). Figure 1C demonstrates that TRPC4 protein expression in U87 cells was up-regulated upon ZIKV infection, while Figure EV2D shows that knocking down TRPC4 in HT22 cells effectively inhibited the production of NS1. Additionally, HC-070 exhibited a protective effect on both U87 and H4 cells infected with ZIKV. The new data are now included in the revised manuscript.

Referee #3: “Zika virus (ZIKV) has been rampant in South America and persistently endemic in regions of Southeast Asia, Africa, and South America in recent years. Infections with ZIKV have been linked to various neurological outcomes, particularly microcephaly in newborns. Dr. Chen and his/her team delved into the neurological manifestations associated with ZIKV infection, pinpointing TRPC4, a calcium channel distinctly implicated in ZIKV infection and its subsequent outcomes. Following studies revealed that ZIKV infection increase the TRPC4 expression via CaMKII-CREB pathway. Notably, antagonists to TPRC4 like HC-070 and the CaMKII inhibitor KN93 effectively curtailed ZIKV proliferation and mitigated related neurological symptoms in in vitro (BHK21 cells), ex vivo (Human brain organoids), and in vivo (ICR suckling mice) models. This study comprehensively evaluated the efficacy of HC-070 and KN93 across various models, presenting potential antiviral therapeutic options for neurological complications linked to ZIKV.

The manuscript was logically organized and well written, but the qualities of some experiments still need to be improved to solidify the conclusion well. Specific recommendations are as follows:"

Response: Thank you for the positive evaluation of our manuscript and valuable suggestions.

Referee #3: "Fig. 1A, the image resolution does not sufficiently facilitate the determination of whether there's a colocalization between elevated TRPC4 expression and ZIKV proteins within cells. In certain ZIKV-infected cells, an apparent increase in TRPC4 expression is not discernible. Moreover, several cells manifesting elevated TRPC4 levels did not appear to be infected by ZIKV. It would be beneficial to provide some images at a higher magnification, which could be incorporated into the Supplementary data. Furthermore, it is advisable to include the MOCK group to showcase the basal expression level of TRPC4 in cells."

Response: We agree that the analysis of immunofluorescence data shown in Figure 1A does not provide sufficient evidence of co-localization between the E protein and TRPC4 channels. The statistical graph presented in Figure 1B represents the average fluorescence intensity observed in panel A, while higher magnification images (EV3 A and B) were included for colocalization analysis. Subsequently, a CO-IP experiment was conducted to explore potential interactions between TRPCs (TRPC1/4/5/6) and ZIKV-E protein overexpression; however, no significant interaction was observed between TRPCs and ZIKV-E protein (EV3 C).

Referee #3: “Fig. 1B, 1D, and corresponding sub-figures, the methodology for calculating grey values remains unclear. Is it based on the fluorescence intensity of all cells captured in the images? This has not been explicitly delineated either in the methods section or the figure legends. Such a quantification technique may not sufficiently capture the evaluate of ZIKV infection or TRPC4 expression changes. Employing grey intensity calculations from Western Blot images (semi-quantification) might offer a more dependable quantification compared to fluorescence evaluation. Alternatively, quantifying the number of infected cells could serve as an alternative measure.”

Response: We greatly appreciate the valuable suggestions provided by the reviewer. Fluorescent images were analyzed using ImageJ software to quantify the mean fluorescence intensity (F), which was subsequently plotted in Figure B. To further validate alterations in TRPC4 protein expression during viral infection, cells were infected with viruses at varying MOI and subjected to Western blot analyses. As illustrated in revised figure 1C, an increase in MOI led to a gradual elevation of TRPC4 protein levels within the cells.

Referee #3: “Fig. 1B, the pooling of data points from both the TRPC4 over-expressed group and the normal infection group for correlation analysis seems inappropriate. It would be more appropriate to analyze these groups separately, or alternatively, consider excluding the TRPC4 over-expressed group from the analysis.”

Response: Thank you for this suggestion. We have removed the TRPC4 overexpression groups and reanalyzed the data. The results of analysis are shown in Figure 1B. The correlation coefficient was found to be 0.83, indicating a strong correlation between the levels of E protein and TRPC4 protein.

Referee #3: “For a deeper understanding of the relationship between TRPC4 and ZIKV infection, exploring varying levels of TRPC4 overexpression in a dose-dependent manner during ZIKV infection could offer valuable insights.”

Response: We carried out additional experiments to investigate the impact of TRPC4 channel overexpression. We transfected varying amounts of TRPC4 plasmid into U87 cells. The Western blot analysis results (revised Figure 1E) demonstrated a positive correlation between NS1 protein levels and the extent of TRPC4 upregulation.

Referee #3: “There's a noticeable discrepancy between Fig. 1C and Fig. 1D. While TRPC4 expression level in Fig. 1C shows a 1.6-fold change, the variance is considerably greater in the Western Blot (Fig 1D). This inconsistency raises questions about the accuracy of the fluorescence determination.”

Response: The extent of TRPC4 upregulation observed in Western blots varied across different trials. However, after conducting independent experiments for 6 times, we found a statistically significant difference in TRPC4 protein levels between the mock and ZIKV infected groups. The representative images with the most significant visible upregulation was selected.

Referee #3: “Fig. 1E, the right panel, comparing the ZIKV RNA levels between the MOCK (un-infected) group and the ZIKV-infected group seems almost redundant or without significant insight.”

Response: ZIKV's RNA was almost undetectable in the Mock group. Mock group had been removed in the revised Figure 1.

Referee #3: “For ZIKV detection, the authors employed a variety of antibodies to identify ZIKV infection, encompassing ZIKV NS1, E, and NS5 proteins. What was the rationale behind selecting different detection targets within a series of similar experiments?”

Response: The antibody used in the immunofluorescence experiment is anti-flavivirus group antigen antibody (clone D1-4G2-4-15, protein E), specifically designed for IF. According to the instructions, this antibody is generally not suitable for WB, so we opted for other available ZIKV protein antibodies such as anti-NS1 or NS5. We have explained this in the revised manuscript. To ensure manuscript uniformity, we conducted additional experiments using an NS1 antibody. Consequently, the revised manuscript incorporates Western blot data utilizing the NS1 antibody.

Referee #3: “For TRPC4 detection, is there a possibility that the antibodies employed to detect TRPC4 may cross-react with other proteins from the TRPC family?”

Response: We conducted a comparative analysis of the amino acid residues corresponding to the immunogens recognized by TRPC4 antibodies, revealing significant

disparities in comparison to other TRPCs (the following picture). Consequently, the likelihood of cross-reactivity with other TRPCs is relatively minimal. To further validate the specificity band of TRPC4, we employed another TRPC4 antibody purchased from Alomone labs (Cat#: ACC-018), which has been confirmed to exhibit negligible cross-reactivity with TRPC1 and TRPC5 (PMID: 31996247 and PMID: 28526717). Similar results were obtained as depicted in the bottom of the following figure.

Referee #3: "There appears to be a discrepancy between the figure labels for Fig 2C and 2D, as they seem to be reversed in contrast to the descriptions provided in the main manuscript."

Response: We apologize for the error. We have corrected it.

Referee #3: "Given the sensitivity of ZIKV particles to certain chemicals, particularly shifts in pH, was there a preliminary assessment to ascertain if HC-070 and KN93 could directly neutralize ZIKV upon co-incubation?"

Response: To investigate the impact of HC-070 and KN93 on viral particles, we conducted an infectivity inhibition assay. The results unequivocally demonstrate that neither of these compounds exhibited a significant effect on ZIKV virulence (revised Figure 5C).

Referee #3: “Fig. 4B, for the column termed as "virus", does it mean the non-treated group?”

Response: Yes, it is. Thank you for noticing this error. We corrected this mistake in the revised manuscript.

Referee #3: “For ZIKV, viral particle entry spans from the initial time point up to 12 hours. The onset of transcription typically starts around 1-2 hours post-entry, followed by the replication process. As deduced from Fig. B, HC-070 appears to have a potent effect during stages III and IV, a moderate impact in stage II. This suggests that the compound primarily functions between the 2hr to 4hr, predominantly affecting the transcription and partially the replication processes. HC-070 showed no co-localization or binding activity to the viral particle, suggesting it has minimal influence on viral entry. Also, it likely has no impact on virus assembly and shows no influence during stage V. Hence, the manuscript's conclusion that "HC-070 acted at almost the entire viral life cycle" doesn't seem precise.”

Response: We agree that HC-070 has minimal influence on viral entry, and we have revised the sentence to improve its clarity.

Referee #3: “The precise mechanism by which HC-070 and the calcium-related pathway take their effects remains to be elucidated. While only TRPC4 is implicated and not other similar TRPC family proteins-despite their relation to calcium regulation-it suggests that TRPC4 might directly interact with certain viral proteins within cells. In vivo Co-IP experiments could be employed to shed light on this aspect.”

Response: The in vivo Co-IP experiments were conducted; however, the obtained results indicate a lack of interaction between ZIKV-E protein and TRPCs (EV3C).

Referee #3: “Line 191, the reference to Fig. EV 2C seems incorrect. Additionally, Line 196, the reference should likely be to Fig. 4C instead of Fig. 4A.”

Response: The error was corrected. Additionally, we have also rearranged the order of the figures.

Referee #3: “Interestingly, the extracellular EGTA treatment suppressed viral production, implying that the viral life cycle depends on calcium influx from the outside. However, this observation might not be directly related to HC-070, as its primary effect is intracellular.”

Response: Although the majority of viruses induce an initial influx of Ca^{2+} and rely on extracellular Ca^{2+} , our Co-IP results indicate minimal interaction between ZIKV-E protein and TRPCs (Figure EV3C), and HC-070 was not able to inhibit the early Ca^{2+} influx induced by ZIKV infection. Therefore, it is likely that the early Ca^{2+} influx induced by ZIKV

infection is not mediated by TRPCs. Thus, the mechanism responsible for mediating the early Ca²⁺ influx during ZIKV infection remains elusive.

Referee #3: “Fig 4F, pCREB appears more prominently detected by antibodies compared to regular CREB. Could this discrepancy be attributed to differences in binding efficiency during Western blot detection?”

Response: The antibodies against CREB and pCREB were obtained from CST (9197s and 9198s). We conducted multiple repeats of this experiment and performed rigorous statistical analysis to address the reviewer's concern. The relevant results are presented in revised Figure 4H.

Referee #3: “Fig. 6E, all mice in the control group were dead by day 15. How were their Racine scores measured for the behavioral experiments on days 15 and 18 post-infection?”

Response: Indeed, by the end of day 15, the last animal in the control group deceased. However, based on the Racine scoring criteria, deceased animals are assigned the highest score of 6.

Referee #3: “Though TRPC4 inhibition has been identified to mitigate seizure symptoms, understanding its correlation with suppressing ZIKV production remains intriguing. Is the reduction of ZIKV integral to the seizure symptom relief or are these two separate narratives?”

Response: This is an excellent point. Indeed, it has been reported that increased expression of TRPC4 in the brain tissue is associated with seizures. We observed an upregulation of TRPC4 expression in the brain tissue of ZIKV-infected mice that was also associated with seizures. The TRPC4 inhibitor decreased viral copies within the brain tissue of ZIKV-infected mice (revised Figure 7C) and mitigated epilepsy symptoms. Thus, it is believed that the relief of seizures can be attributed not only to the functional inhibition of TRPC4 mediated by small molecules but also to the downregulation of TRPC4 induced by viral inhibition.

Referee #3: “The ethics statement should be provided, especially given the use of animals and human tissues in the study.”

Response: The revised manuscript now includes an ethics statement in the methods and materials section. No human tissues were used in this study.

6th Mar 2024

Dear Prof. Obukhov,

Thank you for the submission of your revised manuscript to EMBO Molecular Medicine. We have now heard back from the two referees who we asked to re-evaluate your manuscript. As you will see from the reports below, both referees acknowledge the improvements of the revised manuscript but remain critical particularly regarding the limited mechanistic insight. After a consultation with my colleagues here, we agreed that raised concerns are justified and should be addressed in an additional and final round of major revision.

Please also amend following points:

- Please address all comments suggested by our data editors listed below:

o Figure legends:

1. Please note that a separate 'Data Information' section is required in the legends of figures 1d, f; 2b-d; 3a-d, f-g; 4a-d, g-h, j, l-m; 5b-d; 6b-d; 7a, d-f; EV 2d-e; EV 4b-e.
2. Please define the annotated p values ** in the legend of figure EV 4e as appropriate.
3. Please note that in figure 4m; there is a mismatch between the annotated p values in the figure legend and the annotated p values in the figure file that should be corrected.
4. Please note that information related to n is missing in the legends of figures 1d, f; 2b-d; 3a-d, f-g; 4a-d, g-h, j, l-m; 5b-c; 6b-d; 7a, c-f; EV 1a; EV 2d-e; EV 3d; Ev 4b-e.
5. Although 'n' is provided, please describe the nature of entity for 'n' in the legend of figure 5c.
6. Please note that the scale bar needs to be defined for figures 1a; 4b; EV 2b; EV 3a; EV 4a.
7. Please note that scale bar and its definition are missing for figure EV 2c.
8. Please note that the white boxes are not defined in the legend of figure EV 3a. This needs to be rectified.

- Provide institutional e-mail adress for Ruiyuan Cao.

- Provide "The Paper Explained" and add it to the main manuscript file. For more information please check our "Author Guidelines". <https://www.embopress.org/page/journal/17574684/authorguide#researcharticleguide>

- Please check figure and table callouts in the text. Currently, callouts for Table 1 and Figure EV4 are missing.

- Place table 1 and 2 between main and EV figure legends.

- Rename "Methods" to "Materials and Methods"

- Submit synopsis image as a separate high-resolution 550 px-wide x (250-400)-px high jpeg file and synopsis text as a separate .doc file.

Further consideration of a revision that addresses reviewer's concerns in full will entail an additional round of review.

Acceptance or rejection of the manuscript will depend on the completeness of your responses included in the next, final version of the manuscript. For this reason, and to save you from any frustrations in the end, I would strongly advise against returning an incomplete revision.

We would welcome the submission of a revised version within three months for further consideration. Please let us know if you require longer to complete the revision.

I look forward to receiving your revised manuscript.

Yours sincerely,

Zeljko Durdevic

We require:

2) Individual production quality figure files as .eps, .tif, .jpg (one file per figure). For guidance, download the 'Figure Guide PDF': (<https://www.embopress.org/page/journal/17574684/authorguide#figureformat>).

3) A .docx formatted letter INCLUDING the reviewers' reports and your detailed point-by-point responses to their comments. As part of the EMBO Press transparent editorial process, the point-by-point response is part of the Review Process File (RPF), which will be published alongside your paper.

4) A complete author checklist, which you can download from our author guidelines (<https://www.embopress.org/page/journal/17574684/authorguide#submissionofrevisions>). Please insert information in the checklist that is also reflected in the manuscript. The completed author checklist will also be part of the RPF.

6) It is mandatory to include a 'Data Availability' section after the Materials and Methods. Before submitting your revision, primary datasets produced in this study need to be deposited in an appropriate public database, and the accession numbers and database listed under 'Data Availability'. Please remember to provide a reviewer password if the datasets are not yet public (see <https://www.embopress.org/page/journal/17574684/authorguide#dataavailability>).

13) Author contributions: You will be asked to provide CRediT (Contributor Role Taxonomy) terms in the submission system. These replace a narrative author contribution section in the manuscript.

14) A Conflict of Interest statement should be provided in the main text.

Please also suggest a striking image or visual abstract to illustrate your article as a PNG file 550 px wide x 300-800 px high.

***** Reviewer's comments *****

Referee #2 (Remarks for Author):

It is appreciated that Xingjuan Chen and colleagues performed additional experiments and revised the manuscript; however, this study remains to be fixed before acceptance.

1. The role of TRPC4 in ZIKV infection is unclear, although the general function of TRPC4 in intracellular calcium dynamics has been validated.
 - a. The authors concluded that ZIKV infection increases TRPC4 expression in host cells via the interaction between ZIKV-NS3 protein and CaMKII, enhancing TRPC4-mediated calcium influx. However, TRPC4 inhibitor HC-70 did not suppress the calcium elevation triggered by ZIKV infection (Figure EV3D), suggesting that TRPC4 is not involved in ZIKV-induced calcium influx. The authors should either provide direct evidence that TRPC4 is responsible for the calcium influx by ZIKV or tone it down.
 - b. The authors claim that the ZIKV-NS3 protein enhanced the calcium influx mediated by the TRPC4 channel (Figure 4M and EV4E). However, these data do not exclude the possibility that NS3 is involved in Ca²⁺ influx in parallel with TRPC4. In addition, it is yet to be determined whether the Gd³⁺-induced increase in Ca²⁺ concentration mimics that of ZIKV, and therefore, it cannot be determined whether these experiments are a suitable validation method for analyzing the mechanism of ZIKV-induced Ca²⁺ elevation.
 - c. They describe that overexpression of TRPC4 promotes nuclear localization of DDX3X (Figure 5D), but to accurately assess nuclear localization, the amount of DDX3X localized in the cytoplasm should also be analyzed. Alternatively, since the antibody (11115-1-AP) can be used for immunofluorescence, its nuclear localization should be examined by fluorescence imaging. Moreover, the molecular mechanism by which DDX3 is transported to the nucleus remains to be elucidated because TRPC4 has yet to be shown to be involved in calcium influx by ZIKV.
2. The revised manuscript still needs more detailed descriptions, although the authors claim that they have adequately corrected the lack of explanation in the "Methods" section that this reviewer pointed out. The authors should carefully recheck the manuscript and fix it.
For example, although the authors claimed that they provided detailed descriptions of methods, there needs to be more

description of fluorescence imaging, including descriptions of microscope configuration. The authors should provide clear information to help readers understand the experiments performed.

Referee #3 (Comments on Novelty/Model System for Author):

The authors utilize various model systems, both in vitro and in vivo, which substantially underpin their findings. Additionally, they have included an ethics statement in the revised version of the manuscript, ensuring compliance with ethical standards.

Referee #3 (Remarks for Author):

The authors have addressed several inaccuracies and improved the manuscript significantly, presenting a substantial amount of high-quality data. The efficacy of these chemicals in inhibiting ZIKV has been proven across different model systems. However, a central unresolved issue remains the mechanism of action of HC-070 or KN93. The involvement of the calcium channel related to TRPC4 offers a plausible explanation, yet there are still unresolved debates. These include the variation in chemical concentration acting as either an antagonist or inhibitor of ZIKV infection, the manner in which ZIKV upregulates TRPC4 expression, and how it affects the role of CaMKII, among others. The deeper mechanisms have not been fully explored in this manuscript, potentially diminishing its impact. It is recommended that incorporating a section outlining the weaknesses and limitations of the current mechanism study, along with a concise discussion of these aspects.

Response to the Editor and Reviewers.

We thank the Editor and the Reviewers for valuable suggestions that helped us further improve our manuscript. We include below our point-by-point responses to the Reviewers' comments.

Response to Referee #2

"It is appreciated that Xingjuan Chen and colleagues performed additional experiments and revised the manuscript; however, this study remains to be fixed before acceptance.

1. The role of TRPC4 in ZIKV infection is unclear, although the general function of TRPC4 in intracellular calcium dynamics has been validated.

a. The authors concluded that ZIKV infection increases TRPC4 expression in host cells via the interaction between ZIKV-NS3 protein and CaMKII, enhancing TRPC4-mediated calcium influx. However, TRPC4 inhibitor HC-70 did not suppress the calcium elevation triggered by ZIKV infection (Figure EV3D), suggesting that TRPC4 is not involved in ZIKV-induced calcium influx. The authors should either provide direct evidence that TRPC4 is responsible for the calcium influx by ZIKV or tone it down."

Response: Figure EV 3D shows that ZIKV inoculation leads to intracellular calcium elevation in BHK cells within the first 20 minutes post infection, which is commonly considered as the entry stage of the ZIKV life cycle. At this stage, neither NS3 protein is yet translated nor TRPC4 is upregulated. Therefore, it is not surprising that HC-070 failed to inhibit the ZIKV-induced elevation of intracellular calcium concentration. Consistently, figure 5B shows that HC-070 decreases relative viral RNA copies only during the later stages of infection but not during the entry stage. Additionally, biochemical evidence indicates that there is no interaction between the TRPC4 channel and the viral E protein (Fig EV 3A-C), confirming that TRPC4 does not serve as the receptor for ZIKV entry into cells. These data support the hypothesis that TRPC4 contribution may be insignificant during the early stage of viral entry (discussed in line 407-413).

Conversely, our data indicate that TRPC4-mediated calcium influx may play a role during the later stages of ZIKV infection. To monitor intracellular calcium signals in host cells over an extended period after ZIKV infection, we co-expressed TRPC4, ZIKV-NS3, and a calcium indicator R-GECO cDNAs in BHK cells and then utilized long-term (6 hours) live-cell calcium imaging while maintaining the infected cells in culture medium at 37°C and 5% CO₂. As depicted in Figure EV3E, NS3 increased intracellular calcium levels during long-term live-cell calcium imaging while HC-070 effectively inhibited such increases. These experimental findings further indicate that TRPC4 likely contributes to calcium influx at some later stages following virus entry.

b. The authors claim that the ZIKV-NS3 protein enhanced the calcium influx mediated by the TRPC4 channel (Figure 4M and EV4E). However, these data do not exclude the possibility that NS3 is involved in Ca²⁺ influx in parallel with TRPC4.

Response: Thank you for suggesting important control experiments. We added two additional control groups, NS3 and NS3+H1R, in revised Figures 4M and EV4E. Our new data indicate that ZIKV-NS3 itself did not affect the calcium influx induced by Gd^{3+} or histamine. Furthermore, ZIKV-NS3 did not increase intracellular Ca^{2+} levels in control BHK cells exhibiting low endogenous TRPC4 expression (Fig EV 3E).

In addition, it is yet to be determined whether the Gd^{3+} -induced increase in Ca^{2+} concentration mimics that of ZIKV, and therefore, it cannot be determined whether these experiments are a suitable validation method for analyzing the mechanism of ZIKV-induced Ca^{2+} elevation.

Response : We apologize for any confusion we may have caused. We agree that the experiment with the Gd^{3+} -induced increase in intracellular calcium concentration does not fully mimic the complex relationship between ZIKV and host intracellular calcium changes. The data depicted in Figure 4M are only intended to show that NS3 enhances TRPC4-mediated intracellular calcium increases compared to those observed in TRPC4 solo expressing cells. TRPC4 channels can be activated either by a downstream molecule of the G protein-coupled receptor (GPCR) signaling pathway or by trivalent cations like Gd^{3+} and La^{3+} . Therefore, the TRPC4 channel was activated either by Gd^{3+} (100 μ M) or histamine (1 μ M), an agonist of the H1 GPCR, in this study. The increase in intracellular calcium concentration induced by histamine is considered to be partially mediated by the TRPC4 channel and partly through the IP3R pathway. In contrast, the elevation of intracellular calcium triggered by Gd^{3+} is thought to be facilitated primarily by TRPC4 activation.

c. They describe that overexpression of TRPC4 promotes nuclear localization of DDX3X (Figure 5D), but to accurately assess nuclear localization, the amount of DDX3X localized in the cytoplasm should also be analyzed. Alternatively, since the antibody (11115-1-AP) can be used for immunofluorescence, its nuclear localization should be examined by fluorescence imaging. Moreover, the molecular mechanism by which DDX3 is transported to the nucleus remains to be elucidated because TRPC4 has yet to be shown to be involved in calcium influx by ZIKV.

Response: Thank you for this valuable suggestion. We have performed an immunofluorescence assay using the DDX3X antibody (11115-1-AP) to investigate the nuclear and cytoplasmic localization of DDX3X in infected cells. We found that the mean nuclear/total DDX3X immunofluorescence intensity ratio was increased in ZIKV-infected cells. Regarding the role of TRPC4 in intracellular calcium dynamics in ZIKV infected cells, as we discussed above, our data indicate that TRPC4 is not involved in calcium changes associated with the early stage of the ZIKV life cycle. However, TRPC4 likely contributes to calcium elevation and may underlie the calcium-dependent nuclear translocation of DDX3X during the replication stage of the viral life cycle. Consistently, both HC-070 and EGTA efficiently decreased DDX3X levels in the nuclei of ZIKV-infected

cells.

2. The revised manuscript still needs more detailed descriptions, although the authors claim that they have adequately corrected the lack of explanation in the "Methods" section that this reviewer pointed out. The authors should carefully recheck the manuscript and fix it. For example, although the authors claimed that they provided detailed descriptions of methods, there needs to be more description of fluorescence imaging, including descriptions of microscope configuration. The authors should provide clear information to help readers understand the experiments performed.

Response: We have carefully rechecked our manuscript and expanded the Experimental Methods section. We now provide more detailed information about fluorescence imaging, including microscope configuration (lines 656-660). Additionally, we have provided greater details about the experimental techniques which we employed for monitoring intracellular calcium signals (lines 596-607).

Response to Referee #3:

The authors utilize various model systems, both in vitro and in vivo, which substantially underpin their findings. Additionally, they have included an ethics statement in the revised version of the manuscript, ensuring compliance with ethical standards.

Response: We thank this reviewer for positive comments about our manuscript.

Referee #3 (Remarks for Author):

The authors have addressed several inaccuracies and improved the manuscript significantly, presenting a substantial amount of high-quality data. The efficacy of these chemicals in inhibiting ZIKV has been proven across different model systems. However, a central unresolved issue remains the mechanism of action of HC-070 or KN93. The involvement of the calcium channel related to TRPC4 offers a plausible explanation, yet there are still unresolved debates. These include the variation in chemical concentration acting as either an antagonist or inhibitor of ZIKV infection, the manner in which ZIKV upregulates TRPC4 expression, and how it affects the role of CaMKII, among others. The deeper mechanisms have not been fully explored in this manuscript, potentially diminishing its impact. It is recommended that incorporating a section outlining the weaknesses and limitations of the current mechanism study, along with a concise discussion of these aspects.

Response: Thank you for your insightful comments. Indeed, despite extensive investigation into the molecular mechanism by which ZIKV enhances its replication through up-regulating TRPC4, a comprehensive understanding of ZIKV's impact on the

host calcium signaling system remains not fully understood. We have incorporated a dedicated discussion paragraph on weaknesses and limitations of our study and indicated the need to further elucidate the unresolved intricacies surrounding the deeper mechanisms (lines 482-489).

Regarding the efficacy of HC-070, Figures 3A and 3H show that HC-070 demonstrates a significant antiviral activity starting at approximately 1 μ M and that its maximal inhibitory effect is observed at 10 μ M. Although we cannot rule out the possibility of off-target effects at higher concentrations, the shown dose-response relationship indicates that the ability of HC-070 in inhibiting ZIKV is not far off from the expected range.

We also revised our manuscript to clarify how ZIKV infection upregulates CaMKII and TRPC4. In the revised manuscript, we emphasized the fact that HC-070 and KN-93 inhibited ZIKV RNA production only during the replication stage of ZIKV life cycle. Conversely, neither HC-070 nor KN-93 exhibited any significant effect on ZIKV virulence (Figure 5 C). Thus, the effects of these two inhibitors were specific.

18th Jun 2024

Dear Prof. Obukhov,

Thank you for the submission of your revised manuscript to EMBO Molecular Medicine. I am pleased to inform you that we will be able to accept your manuscript pending the following final amendments:

- 1) Please address the referee's minor concern and apply appropriate statistical test including multivariate analysis of variance as suggested.
- 2) Figures: We note that in Figure 4I values for CaMKII and NS1 are identical. Also, during our standard source data analysis we note that the values of CaMKII and NS1 for Figure 4I are duplicated. We would like to clarify these issues before we proceed with publication of your manuscript. We kindly invite you to check attached source data excel file with identified duplicated values that are color labeled and clarify the cause of these duplications and potentially also amend the figure 4I.
- 3) In addition to the number and nature of replicates in the figure legends please also indicate exact p = values, not a range, along with the statistical test used. To keep the figures "clear" some authors found providing an Appendix table Sx with all exact p -values preferable. You are welcome to do this if you want to.
- 4) Please include structured Methods section that includes a Reagents and Tools Table followed by a Methods and Protocols section. File EV1 seems to be a detailed protocol in table format, please add it to the "Appendix" and rename tables to "Appendix Table S1" etc. and update the callouts in the text. Please check "Author Guidelines" for more information and to download table templates. <https://www.embopress.org/page/journal/17574684/authorguide#structuredmethods>
An example of a Method paper with Structured Methods can be found here:
<https://www.embopress.org/doi/full/10.1038/s44320-024-00037-6#sec-4>
- 5) The Paper Explained: Please change subheadings to "Problem", "Results" and "Impact".
- 6) For more information: This space should be used to list relevant web links for further consultation by our readers. Could you identify some relevant ones and provide such information as well? Some examples are patient associations, relevant databases, OMIM/proteins/genes links, author's websites, etc...
- 7) As part of the EMBO Publications transparent editorial process initiative (see our Editorial at <http://embomolmed.embopress.org/content/2/9/329>), EMBO Molecular Medicine will publish online a Review Process File (RPF) to accompany accepted manuscripts. This file will be published in conjunction with your paper and will include the anonymous referee reports, your point-by-point response and all pertinent correspondence relating to the manuscript. Let us know whether you agree with the publication of the RPF and as here, if you want to remove or not any figures from it prior to publication. Please note that the Authors checklist will be published at the end of the RPF.
- 8) Please provide a point-by-point letter INCLUDING my comments as well as the reviewer's reports and your detailed responses (as Word file).

I look forward to reading a new revised version of your manuscript as soon as possible.

Yours sincerely,

Zeljko Durdevic

*** Instructions to submit your revised manuscript ***

To submit your manuscript, please follow this link:

<https://embomolmed.msubmit.net/cgi-bin/main.plex>

- 1) a .docx formatted version of the manuscript text (including Figure legends and tables)
 - 2) Separate figure files*
 - 3) supplemental information as Expanded View and/or Appendix. Please carefully check the authors guidelines for formatting Expanded view and Appendix figures and tables at <https://www.embopress.org/page/journal/17574684/authorguide#expandedview>
 - 4) a letter INCLUDING the reviewer's reports and your detailed responses to their comments (as Word file).
 - 5) The paper explained: EMBO Molecular Medicine articles are accompanied by a summary of the articles to emphasize the major findings in the paper and their medical implications for the non-specialist reader. Please provide a draft summary of your article highlighting
 - the medical issue you are addressing,
 - the results obtained and
 - their clinical impact.This may be edited to ensure that readers understand the significance and context of the research. Please refer to any of our published articles for an example.
 - 6) For more information: There is space at the end of each article to list relevant web links for further consultation by our readers. Could you identify some relevant ones and provide such information as well? Some examples are patient associations, relevant databases, OMIM/proteins/genes links, author's websites, etc...
 - 7) Author contributions: the contribution of every author must be detailed in a separate section.
 - 8) EMBO Molecular Medicine now requires a complete author checklist (<https://www.embopress.org/page/journal/17574684/authorguide>) to be submitted with all revised manuscripts. Please use the checklist as guideline for the sort of information we need WITHIN the manuscript. The checklist should only be filled with page numbers where the information can be found. This is particularly important for animal reporting, antibody dilutions (missing) and exact values and n that should be indicated instead of a range.
 - 9) Every published paper now includes a 'Synopsis' to further enhance discoverability. Synopses are displayed on the journal webpage and are freely accessible to all readers. They include a short stand first (maximum of 300 characters, including space) as well as 2-5 one sentence bullet points that summarise the paper. Please write the bullet points to summarise the key NEW findings. They should be designed to be complementary to the abstract - i.e. not repeat the same text. We encourage inclusion of key acronyms and quantitative information (maximum of 30 words / bullet point). Please use the passive voice. Please attach these in a separate file or send them by email, we will incorporate them accordingly.
- You are also welcome to suggest a striking image or visual abstract to illustrate your article. If you do please provide a jpeg file 550 px-wide x 300-800px high.
- 10) A Conflict of Interest statement should be provided in the main text
 - 11) Please note that we now mandate that all corresponding authors list an ORCID digital identifier. This takes <90 seconds to complete. We encourage all authors to supply an ORCID identifier, which will be linked to their name for unambiguous name identification.

Currently, our records indicate that the ORCID for your account is 0000-0002-3862-6004.

Link Not Available

Photos 400-800 DPI

*Additional important information regarding figures and illustrations can be found at <https://bit.ly/EMBOPressFigurePreparationGuideline>. See also figure legend preparation guidelines: <https://www.embopress.org/page/journal/17574684/authorguide#figureformat>

***** Reviewer's comments *****

Referee #2 (Remarks for Author):

The current version of the manuscript has been improved significantly by the additional experiments and appropriate revisions according to the reviewers' comments. Therefore, this study should be published after the remaining issue is addressed. Statistical analysis in Figure EV3G (Dunnett's post hoc test) is inappropriate. The authors should perform appropriate tests, including multivariate analysis of variance (MANOVA) with Bonferroni correction, for data over time.

Response to reviewer

The current version of the manuscript has been improved significantly by the additional experiments and appropriate revisions according to the reviewers' comments. Therefore, this study should be published after the remaining issue is addressed.

Response: We appreciate your positive comments regarding our revised manuscript.

Statistical analysis in Figure EV3G (Dunnett's post hoc test) is inappropriate. The authors should perform appropriate tests, including multivariate analysis of variance (MANOVA) with Bonferroni correction, for data over time.

Response: Thank you for your suggestion regarding the statistical analysis in Figure EV3G. The data in EV3G was re-analyzed using MANOVA with Bonferroni correction in the revised manuscript, as suggested. The statistical conclusion remained unchanged. Thank you once again for your insightful comments and for helping us improve the rigor of our study.

Response to editor

Figures: We note that in Figure 4I values for CaMKII and NS1 are identical. Also, during our standard source data analysis we note that the values of CaMKII and NS1 for Figure 4I are duplicated. We would like to clarify these issues before we proceed with publication of your manuscript. We kindly invite you to check attached source data excel file with identified duplicated values that are color labeled and clarify the cause of these duplications and potentially also amend the figure 4I.

Response: We sincerely apologize for this oversight. We have thoroughly re-examined our source data and discovered that the values for CaMKII and NS1 in Figure 4I were inadvertently duplicated during the process of copying and pasting from raw data. We have revised both the source data and the figure accordingly. The revised Figure 4I, along with the corrected source data, will be uploaded again.

1st Jul 2024

Dear Prof. Obukhov,

We are pleased to inform you that your manuscript is accepted for publication and is now being sent to our publisher to be included in the next available issue of EMBO Molecular Medicine.
